# Pharmacological dimerization and activation of the exchange factor eIF2B antagonizes the integrated stress response

Carmela Sidrauski[1,2][\*][†][‡], Jordan C Tsai[1,2][†], Martin Kampmann[2,3], Brian R Hearn[4,5], Punitha Vedantham[4,5], Priyadarshini Jaishankar[4,5], Masaaki Sokabe[6], Aaron S Mendez[2,3], Billy W Newton[7], Edward L Tang[7,8], Erik Verschueren[7], Jeffrey R Johnson[7,8], Nevan J Krogan[7,8], Christopher S Fraser[6], Jonathan S Weissman[2,3], Adam R Renslo[4,5], Peter Walter[1,2][\*]

[1]Department of Biochemistry and Biophysics, University of California, San Francisco, San Francisco, United States; [2]Howard Hughes Medical Institution, University of California, San Francisco, San Francisco, United States; [3]Department of Cellular and Molecular Pharmacology, University of California, San Francisco, San Francisco, United States; [4]Department of Pharmaceutical Chemistry, University of California, San Francisco, San Francisco, United States; [5]Small Molecule Discovery Center, University of California, San Francisco, San Francisco, United States; [6]Department of Molecular and Cellular Biology, College of Biological Sciences, University of California, Davis, Davis, United States; [7]QB3, California Institute for Quantitative Biosciences, University of California, San Francisco, San Francisco, United States; [8]Gladstone Institutes, San Francisco, United States

*For correspondence: carmelas@me.com (CS); peter@walterlab.ucsf.edu (PW)

[†]These authors contributed equally to this work

Present address: [‡]Calico LLC, South San Francisco, United States

**Abstract** The general translation initiation factor eIF2 is a major translational control point. Multiple signaling pathways in the integrated stress response phosphorylate eIF2 serine-51, inhibiting nucleotide exchange by eIF2B. ISRIB, a potent drug-like small molecule, renders cells insensitive to eIF2α phosphorylation and enhances cognitive function in rodents by blocking long-term depression. ISRIB was identified in a phenotypic cell-based screen, and its mechanism of action remained unknown. We now report that ISRIB is an activator of eIF2B. Our reporter-based shRNA screen revealed an eIF2B requirement for ISRIB activity. Our results define ISRIB as a symmetric molecule, show ISRIB-mediated stabilization of activated eIF2B dimers, and suggest that eIF2B4 (δ-subunit) contributes to the ISRIB binding site. We also developed new ISRIB analogs, improving its EC$_{50}$ to 600 pM in cell culture. By modulating eIF2B function, ISRIB promises to be an invaluable tool in proof-of-principle studies aiming to ameliorate cognitive defects resulting from neurodegenerative diseases.

## Introduction

In the integrated stress response (ISR), phosphorylation of the α-subunit of the eukaryotic translation initiation factor eIF2 (eIF2α-P) at serine-51 acts as a major regulatory step that controls the rate of translation initiation. Four distinct eIF2α kinases can catalyze phosphorylation at this single residue, each acting in response to different cellular stress conditions: PERK senses accumulation of unfolded polypeptides in the lumen of the endoplasmic reticulum (ER), GCN2 responds to amino acid starvation

**eLife digest** Proteins are often described as life's 'workhorse' molecules, and cells must be able to build new proteins to stay alive. This ability is also vital for storing new memories. A protein called eIF2 carries out a critical step in the process that cells use to make proteins; and a decrease in the activity of eIF2 has been linked with memory loss in diseases such as Parkinson's and Alzheimer's disease.

When a cell experiences stressful conditions—such as virus infection or starvation—it triggers the 'integrated stress response'. This response helps the cell conserve its resources and take corrective steps to restore its normal working conditions. As part of the integrated stress response, an enzyme adds a phosphate group onto eIF2. The 'phosphorylated' eIF2 blocks protein production, which causes the cell to make fewer proteins. In 2013, researchers revealed that a small drug-like molecule, called ISRIB, could prevent this decline in protein production following eIF2 phosphorylation; and when ISRIB was administered to mice and rats, it enhanced their long-term memories. However, this early work did not identify the molecule that is targeted by ISRIB.

Now Sidrauski, Tsai et al.—including many of researchers involved in the 2013 work—have used a combination of techniques including genetics, chemistry and biochemistry to reveal the target of ISRIB. The experiments show that ISRIB's molecular target is a protein complex called eIF2B. Artificially reducing the production of eIF2B made cells resistant to the effects of ISRIB. The eIF2B protein normally works to activate eIF2; Sidrauski, Tsai et al. observed that ISRIB boosts the activity of eIF2B and renders it insensitive to blockage by phosphorylated eIF2. This in turn increases protein production in the cell.

But how does ISRIB activate eIF2B? It was known that two copies of eIF2B can bind to each other; and Sidrauski, Tsai et al. found that ISRIB acts by stabilizing these larger protein complexes that are more active and less sensitive to inhibition by phosphorylated eIF2. Finally, in further experiments, new versions of ISRIB were synthesized that are ten-times as active as the original molecule inside cells.

Importantly, the discovery that eIF2B is the molecular target for ISRIB has recently been independently validated by other researchers, and it looks promising that this discovery will guide future efforts to develop clinically useful drugs to treat memory disorders.

and UV-light, PKR responds to viral infection, and HRI responds to heme deficiency. Their convergence on the same molecular event leads to a reduction in overall protein synthesis. Concomitant with a decrease in new protein synthesis, preferential translation of a small subset of mRNAs that contain small upstream open reading frames (uORFs) in their 5′ untranslated region is induced (*Harding et al., 2003*; *Wek et al., 2006*). ISR-translational targets include the well-known mammalian ATF4 (Activating Transcription Factor 4) and CHOP (a pro-apoptotic transcription factor) (*Harding et al., 2000*; *Vattem and Wek, 2004*; *Palam et al., 2011*). ATF4 regulates genes involved in metabolism and nutrient uptake and was shown to have a cytoprotective role upon stress in many cellular contexts (*Ye et al., 2010*). ATF4 is also a negative regulator of 'memory genes' and its preferential translation in neurites can transmit a neurodegenerative signal in neurons (*Chen et al., 2003*; *Baleriola et al., 2014*). ISR activation leads to preferential translation of key regulatory molecules and thus its level and duration of induction must be tightly regulated. Cells ensure that the effects of eIF2α-P are transient by also activating a negative feedback loop. This is accomplished by GADD34 induction, which encodes the regulatory subunit of the eIF2α phosphatase (*Lee et al., 2009*). GADD34 induction leads to a reduction of eIF2α-P, allowing cells to restore translation (*Novoa et al., 2001*).

eIF2 is a trimeric complex (comprised of α, β and γ subunits) that binds to both GTP and the initiator methionyl tRNA (Met-tRNA$_i$) to form a ternary complex (eIF2•GTP•Met-tRNA$_i$). After engaging the 40S ribosomal subunit at an AUG start codon recognized by Met-tRNA$_i$, GTP is hydrolyzed by the GTPase activating protein (GAP) eIF5, and the 60S ribosomal subunit joins to form a complete 80S ribosome ready for polypeptide elongation. eIF2•GDP is released, and eIF2 must then be reloaded with GTP to enter another round of ternary complex formation (*Hinnebusch and Lorsch, 2012*). In addition to being a GAP for eIF2, eIF5 is also a GDP dissociation inhibitor that prevents GDP release from eIF2 (*Jennings and Pavitt, 2015*). The exchange of GDP with GTP in eIF2 is catalyzed by its dedicated

guanine nucleotide exchange factor (GEF) eIF2B, which has the dual function of catalyzing the release of both eIF5 and GDP (*Jennings et al., 2013*). eIF2B is a complex molecular machine, composed of five different subunits, eIF2B1 through eIF2B5, also called the α, β, γ, δ, and ε subunits. eIF2B5 catalyzes the GDP/GTP exchange reaction and, together with a partially homologous subunit eIF2B3, constitutes the 'catalytic core' (*Williams et al., 2001*). The three remaining subunits (eIF2B1, eIF2B2, and eIF2B4) are also highly homologous to one another and form a 'regulatory sub-complex' that provides binding sites for eIF2B's substrate eIF2 (*Dev et al., 2010*). When phosphorylated on Ser-51, eIF2α-P dissociates more slowly from the eIF2B regulatory sub-complex and locks eIF2B into an inactive state (*Krishnamoorthy et al., 2001*). Phosphorylation thus renders eIF2 an inhibitor of its own GEF. Because eIF2 is more abundant than eIF2B, a small amount of eIF2α-P is sufficient to sequester a large proportion of available eIF2B, leading to a substantial reduction in overall protein synthesis.

Using a cell-based high-throughput screen, we recently identified a small molecule, ISRIB (for i̱ntegrated s̱tress ṟesponse i̱nhiḇitor) that renders cells resistant to the inhibitory effects of eIF2α-P. ISRIB, the only *bona fide* ISR inhibitor identified to date, is a highly potent compound ($EC_{50} = 5$ nM in cells) and has good pharmacokinetic properties (*Sidrauski et al., 2013*). In agreement with the phenotype of genetically modified mice having reduced eIF2α-P, we showed that treatment with ISRIB enhances memory consolidation in rodents. Moreover, ISRIB comprehensively and selectively blocked the effects of eIF2α phosphorylation on mRNA translation and triggered rapid stress granule disassembly (*Sidrauski et al., 2015*). To date, the molecular target of ISRIB is not known. The fast kinetics of action of ISRIB and the remarkable specificity of its effects in response to eIF2α phosphorylation strongly suggested that its target is a factor that closely interacts with the eIF2 translation initiation complex. The existence of eIF2B mutations in yeast that, like ISRIB, render cells resistant to eIF2α-P led us to propose that eIF2B was a likely target of this small molecule (*Sidrauski et al., 2013*). Here, we draw on clues from two independent approaches, an unbiased genetic screen and structure/activity analyses of ISRIB, to converge on the hypothesis that the mammalian eIF2B complex indeed is the molecular target of ISRIB. We demonstrate that a symmetric ISRIB molecule induces or stabilizes eIF2B dimerization, increasing its GEF activity and desensitizing it to inhibition by eIF2-P. Thus ISRIB directly modulates the central regulator in the ISR.

## Results

### Knockdown of eIF2B renders cells resistant to ISRIB

To identify the molecular target of ISRIB, we conducted a genetic screen for genes whose knockdown modulated the sensitivity of cells to the drug. Using this strategy, we were previously able to pinpoint the molecular targets of cytotoxic compounds and to delineate their mechanism of action (*Matheny et al., 2013*; *Julien et al., 2014*). Here, we conducted a reporter-based screen using a sub-library of our next-generation shRNA library targeting 2933 genes involved in aspects of proteostasis. This focused library targets each protein-coding gene with ~25 independent shRNAs and contains a large set (>1000) of negative-control shRNAs. We have previously shown that the use of such libraries and analysis using a rigorous statistical framework generates robust results from forward genetic screens (*Bassik et al., 2013*; *Kampmann et al., 2013*). We screened the shRNA library in a K562 cell line expressing an uORF-ATF4-venus reporter (*Figure 1A*), similar to the translational reporters that we and others previously used to measure activation of the ISR. In cells bearing this reporter, the venus fluorescent protein is translationally induced upon eIF2α phosphorylation. We chose the K562 cell line for the screen because these cells are non-adherent and allow for efficient fluorescence-activated cell sorting (FACS). Treatment with thapsigargin (Tg), an ER stress inducer that inhibits the ER-localized $Ca^{2+}$-ATPase, resulted in a sixfold increase in mean fluorescence intensity and, as expected, ISRIB substantially reduced induction of the reporter (*Figure 1B*). As a first step in the screen, we transduced the reporter cell line with the library and selected shRNA-expressing cells. We next divided the population and induced ER stress with Tg in the presence or absence of ISRIB. To optimize the dynamic range of the screen and to focus on early translational effects elicited by eIF2α phosphorylation, we incubated cells for 7 hr, at which time full induction of the reporter was reached. To identify genes whose knockdown resulted in either enhanced or reduced sensitivity to ISRIB, we used a concentration of drug corresponding to the $EC_{50}$ (15 nM) in this cell type. Cells from each subpopulation (Tg-treated and Tg + ISRIB-treated) were then FACS-sorted to isolate the third of the

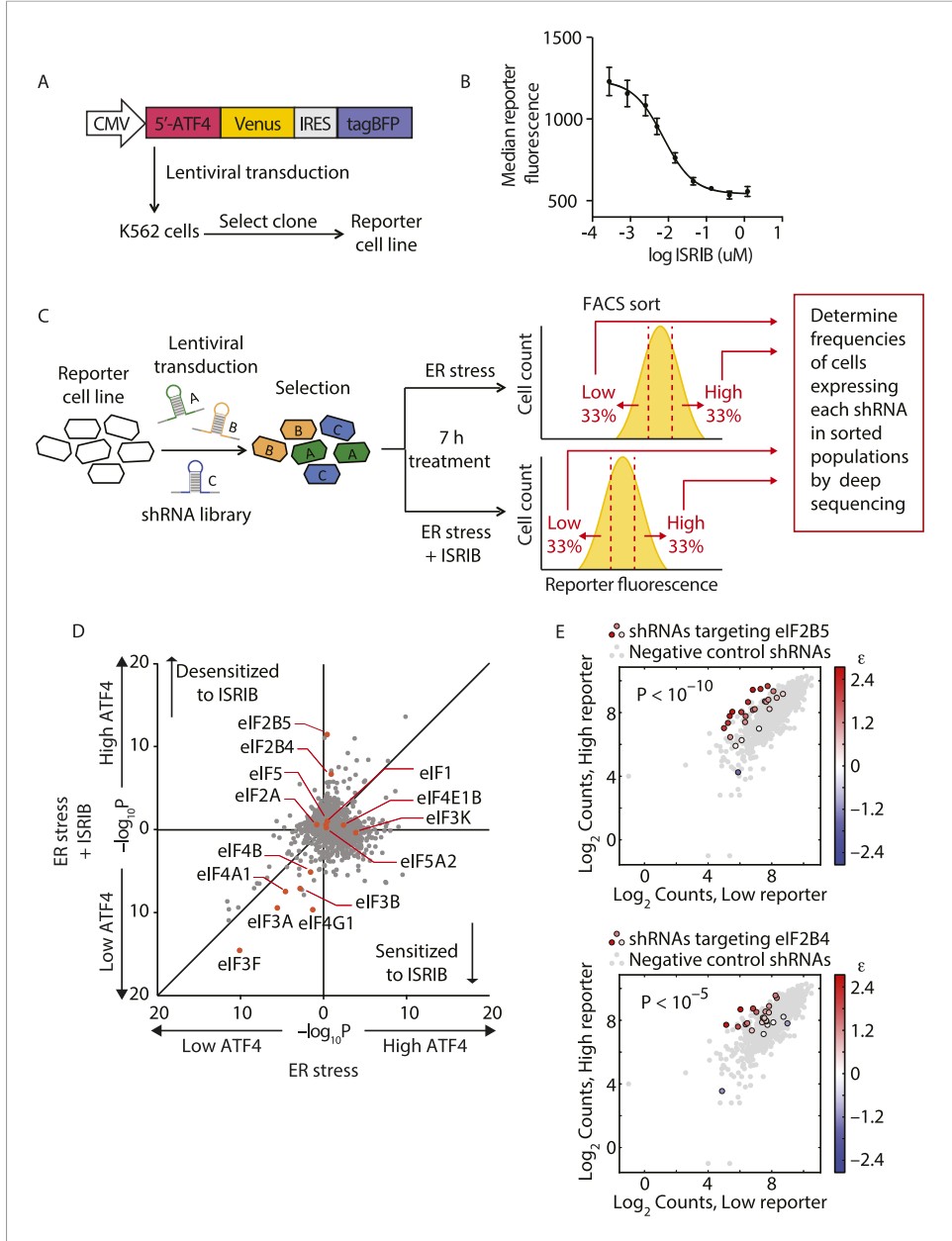

**Figure 1**. Knockdown of eIF2B subunits renders cells more resistant to ISRIB. (**A**) Schematic representation of the ATF4-venus reporter used for the screen. The 5′ end of the human ATF4 mRNA up to the start codon of the ATF4-encoding ORF was fused to venus, followed by the EMCV internal ribosomal entry site (IRES) and BFP and inserted into a lentiviral system. (**B**) ISRIB reduces activation of the ATF4-venus reporter. K562 cells were incubated with Tg (300 nM) for 6 hr in the presence of different concentrations of ISRIB. Reporter fluorescence was measured by flow cytometry and median values were plotted (N = 3, ± SD). (**C**) Schematic of the shRNA screen aimed to identify the target ISRIB. K562 cells expressing the screening reporter were transduced with a pooled shRNA library and transduced cells were selected. The population was then divided into two and either treated with Tg (ER stress) or Tg + ISRIB (ER stress + ISRIB) for 7 hr. Cells were sorted based on their fluorescence (venus) intensity into three bins and the third of the population with the Low and High-reporter levels were collected. Note that the ER stress + ISRIB population had a lower overall fluorescence intensity (median) as ISRIB partially blocks induction of the reporter when added at a concentration corresponding to its $EC_{50}$ in these cells (15 nM). DNA was extracted from the sorted subpopulations for each treatment and shRNA-encoding cassettes were PCR-amplified and subjected to deep sequencing to determine their frequency. (**D**) Effect of knockdown of individual genes in the proteostasis library on reporter expression upon ISR induction in the presence and absence of ISRIB. Gene p values for enrichment and depletion were compared between the ER stress (x-axis) vs the ER stress + ISRIB (y-axis) experiments. For each gene,

*Figure 1. continued on next page*

*Figure 1. Continued*

a p value was calculated by comparing the distribution of $\log_2$ enrichment values for the 25 shRNAs targeting the gene to the negative control shRNAs. (**E**) The $\log_2$ counts for eIF2B5 (top panel) or eIF2B4 (bottom panel) targeting shRNAs in the High-reporter population (x-axis) vs the Low-reporter population (y-axis) was plotted and color coded based on the $\log_2$ enrichment as depicted in the side bar. Red colors indicate a shift towards higher reporter levels, blue colors shifts towards lower reporter levels. Negative control shRNAs in the library are colored grey.
The following source data are available for figure 1:

**Source data 1**. Sequence of the reporter utilized in the shRNA screen.
**Source data 2**. Gene p values for the High and Low reporter populations.

population with the lowest reporter expression and the third of the population with the highest reporter expression (see schematic in *Figure 1C*). To quantify frequencies of cells expressing each shRNA, we isolated genomic DNA from the sorted populations and then PCR-amplified, purified and analyzed by deep-sequencing the shRNA-encoding cassettes. To determine the enrichment or depletion of each shRNA, we compared its frequency in the Low and High reporter populations. For each gene, we calculated a p value by comparing the distribution of $\log_2$ enrichment for the 25 shRNAs targeting the gene to the negative control shRNAs. We then plotted p values for each gene determined in ER stress-induced cells in the absence (x-axis) vs the presence (y-axis) of ISRIB (*Figure 1D*).

The data shown in *Figure 1D* revealed that knockdown of the majority of the genes in the library did not change the expression of the reporter upon either treatment and thus congregated in the center of the plot. By contrast, knockdown of genes that changed the expression of the reporter to the same degree in both treatments localized to the diagonal. We focused our analysis on genes that when knocked-down in the presence of ISRIB, affected the expression of the reporter selectively. In this plot these genes are displaced along the y-axis and encode proteins whose reduced expression modulates the cells' sensitivity to ISRIB. Knockdown of genes that confer resistance to ISRIB lie above the diagonal, while knockdown of genes that confer hypersensitivity to ISRIB lie below it.

Of particular interest was the pronounced effect of the knockdown of (i) two subunits of eIF2B, eIF2B4 and eIF2B5, that significantly reduced the sensitivity (p < $1.4 \cdot 10^{-6}$ and p < $2.4 \cdot 10^{-11}$, respectively) and (ii) eIF4G1 that significantly enhanced the sensitivity (p < $3.4 \cdot 10^{-10}$) of cells to ISRIB, each without affecting induction of the reporter (i.e., no displacement along the x-axis). Individual shRNAs targeting either eIF2B4 or eIF2B5 were enriched in the High reporter population of the ISRIB-treated sample and stood out from the negative control shRNA population (*Figure 1E*). Knockdown of other translation initiation factors (highlighted in *Figure 1D*) revealed no effects on ISRIB sensitivity (locating close to the diagonal of the plot). Based on these data and the fact that eIF2α-P is a direct inhibitor of eIF2B, we postulated that eIF2B is a promising candidate target of ISRIB. Moreover, the data suggest that ISRIB acts as an activator of eIF2B: when eIF2B levels are reduced, cells become resistant to the effects of ISRIB when there is a lower supply of molecules that can be activated.

## Structure-activity relationship of ISRIB suggests a twofold symmetric target

Structure-activity studies of synthetic ISRIB analogs provided further clues as to the nature of its molecular target in cells. Of particular note is that the progenitor member of this class (ISRIB, also denoted herein as ISRIB-A1, *Figure 2A*) exhibits twofold rotational symmetry and is bisected longitudinally by a mirror plane. The molecule is thus achiral but can exist as either *cis* or *trans* diastereomers, depending on the relative orientation of the side chains at positions 1 and 4 of the cyclohexane ring (*Figure 2A*, ISRIB-A1 and ISRIB-A2). We previously showed in cell-based assays that the *trans*-isomer (ISRIB-A1, $EC_{50}$ = 5 nM) is > 100-fold more potent than the *cis*-isomer (ISRIB-A2, $EC_{50}$ > 600 nM). This indicated a preference for an extended binding conformation, with both side chains adopting an equatorial position, as would be expected in the preferred chair conformation of the *trans* diastereomer (ISRIB-A1) (*Sidrauski et al., 2013*). By contrast, the *cis* diastereomer ISRIB-A2 would need to adopt a higher-energy boat-like conformation to project both side chains in pseudo-equatorial orientations. Further structure-activity studies revealed that a 1,4-phenyl spacer could reasonably substitute for 1,4-cyclohexyl, although a 10-fold loss in potency was observed (ISRIB-A7,

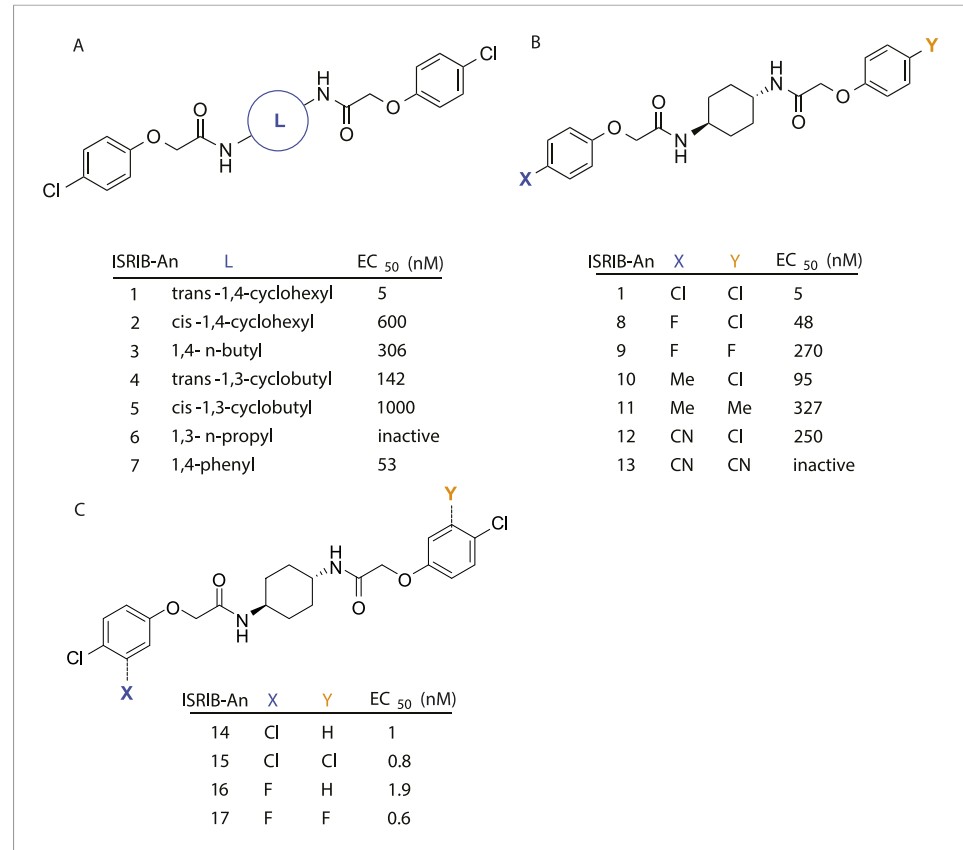

**Figure 2**. SAR analyses suggest ISRIB interacts with a twofold symmetric target. (**A**) ISRIB analogs bearing various likers (L) between the pendant side chains and their corresponding $EC_{50}$ values. (**B**) Sequential replacement of the *para*-chloro substituent (X and Y) with F, Me, or CN on the distal aromatic rings has unfavorable and additive effects on potency. (**C**) Sequential addition of a *meta*-substituent (X and Y) on the distal aromatic rings had favorable and additive effects on potency. Dose response curves of the different ISRIB analogs are shown in ***Figure 2—figure supplement 1***.

The following figure supplement is available for figure 2:

**Figure supplement 1**. Activation of the ATF4 luciferase reporter in HEK293T cells was measured.

$EC_{50} = 53$ nM). Replacement of the 1,4-cyclohexyl ring with *cis* or *trans*-1,3-cyclobutyl spacers resulted in a more dramatic loss of potency (ISRIB-A4, $EC_{50} = 142$ nM; ISRIB-A5, $EC_{50} = 1000$ nM), indicating that the distance between the distal aromatic rings in ISRIB analogs is as important as their positioning in space. This distance dependence was also observed in analogs with acyclic spacers (e.g., ISRIB-A3 and ISRIB-A6). Thus, the *n*-butyl linker in ISRIB-A3 (maintaining the spacing of ISRIB-A1) was better tolerated than the shorter *n*-propyl linker in ISRIB-A6, an analog without measurable activity. The 60-fold reduction in the potency of ISRIB-A3 as compared to ISRIB-A1 can be explained by the increased flexibility of the *n*-butyl chain, resulting in a higher entropic cost associated with adopting the conformation required for binding.

Extensive structure-activity relationship (SAR) studies were also carried out on the distal aryl substituents. Overall, we found that the SAR was consistent with the idea that ISRIB analogs bind across a symmetrical interface. Thus, sequential modification of one and then both side chains in ISRIB analogs was additive, both for favorable modifications and for unfavorable modifications. For example, a *para*-chloro substituent was found to be optimal in ISRIB analogs. Replacing one or both *para*-chloro substituents with fluoro, methyl, or cyano groups led to predictable deterioration of potencies, with the doubly modified analogs least potent in every case (***Figure 2B***, compare ISRIB-A8 with A9, ISRIB-A10 with A11 and ISRIB-A12 with A13). Conversely, the addition of a *meta*-chloro or

*meta*-fluoro substituent enhanced the potency of ISRIB analogs, and introducing such modifications on both side chains produced the most potent analogs (*Figure 2C*, compare ISRIB-A14 with A15, ISRIB-A16 with A17). Among these more potent analogs is ISRIB-A17, which is nearly 10-fold more potent than ISRIB-A1, lowering the $EC_{50}$ into the picomolar range. A full account of our SAR studies will be provided elsewhere but the data presented here demonstrate that the electronics of the phenoxy substituents are important drivers of potency and support the notion that the two halves of ISRIB analogs are engaged in similar recognition events with the target. The most plausible explanation of these findings is that the functional twofold symmetry of ISRIB reflect a target that is likewise twofold symmetric. Taken together, the results obtained by the shRNA screen described above and the recent discovery of eIF2B dimers suggest that ISRIB may act by directly binding to eIF2B at a twofold symmetric interface that stabilizes it as a dimer (*Gordiyenko et al., 2014*; *Wortham et al., 2014*).

## ISRIB promotes dimerization of eIF2B in cells

To test directly whether ISRIB induces or stabilizes the dimeric form of eIF2B, we treated cells with or without ISRIB. We prepared extracts in a high-salt buffer to dissociate eIF2B from its substrate eIF2 and analyzed the lysates by velocity sedimentation on sucrose gradients. In the absence of ISRIB, eIF2B (as detected by immunoblotting with antibodies against eIF2B4 and eIF2B5) migrated predominantly in fractions 3–6 in the gradient, consistent a combined molecular mass of four of its subunits (225 kDa). In the high-salt buffer used, the eIF2B complex lacked the eIF2B1 subunit, which was found predominantly in fractions 1–3 of the gradient. By contrast, when cells were treated with ISRIB, we observed a substantial shift in sedimentation towards a higher molecular mass (predominantly found in fractions 5–8), demonstrating a substantial increase in complex size. By comparing the relative mobility of eIF2B4 and eIF2B5 to that of a background band (marked with a red asterisk in the upper panel of *Figure 3*), the shift in size of eIF2B is easily appreciated. The magnitude of the shift is consistent with a doubling in the molecular mass of the complex. Interestingly, in extracts from ISRIB-treated cells, eIF2B1 also shifted to the heavier fractions, suggesting that its association with the rest of the complex was stabilized. In contrast to the eIF2B subunits, we did not observe a shift in eIF3a or eIF2α. These data strongly support the notion that ISRIB induces the formation of a stable eIF2B dimer.

To determine if eIF2B's ostensible increase in molecular mass was due to dimerization of a complete eIF2B complex, we used mass spectrometry to validate the shift of all of its five subunits. To this end, we treated cells with ISRIB or with an inactive analog ('ISRIB^inact' [ISRIB-A18], *Figure 3—figure supplement 1*) and subjected extracts to fractionation on sucrose gradients. We used ISRIB^inact to control for non-specific hydrophobic interactions of ISRIB with proteins in the extract. We determined the complete protein composition in the fractions in which eIF2B peaked in the presence of ISRIB (fractions 6–9, *Figure 3—figure supplement 2*) by mass spectrometry. This analysis revealed a significant ISRIB-dependent enrichment of all five eIF2B subunits (*Figure 3B*). Notably, eIF2B subunits in ISRIB samples exhibited a characteristic profile in which all subunits collectively peaked in fraction 7. By contrast eIF2B subunits in ISRIB^inact samples were most abundant in fraction 6 and trailed further into the gradient. As expected, two other large protein complexes, the proteasome (*Figure 3B*; data shown for subunit PSMD1) and eIF3 (*Figure 3B*; data shown for subunit eIF3A), showed no displacement upon ISRIB treatment.

Because the mass spectrometric analysis of the gradient was performed with a non-targeted method, it allowed us to ask whether additional proteins would associate with eIF2B potentially contributing to the shift in size. To address this question, we correlated the intensity profiles of all other proteins identified through the analyzed fractions to the sedimentation profile exhibited by a representative subunit, eIF2B4. We plotted the correlation coefficient (R-value) for each comparison. We were excited to find that all eIF2B subunits (eIF2B1, eIF2B2, eIF2B3, eIF2B5) stood out as most strongly correlated to eIF2B4, all exhibiting correlation coefficients (R-values) > 0.98 (*Figure 3C*), strongly indicating that the increase in molecular mass of eIF2B upon ISRIB addition indeed resulted from eIF2B dimerization. Moreover, these analyses strongly support the notion that eIF2B forms a complete complex upon ISRIB treatment.

## ISRIB enhances the thermo-stability of eIF2B4

To identify the subunit of eIF2B targeted by ISRIB, we monitored drug-target engagement, utilizing a cellular extract thermal shift assay (CETSA) (*Martinez Molina et al., 2013*). This

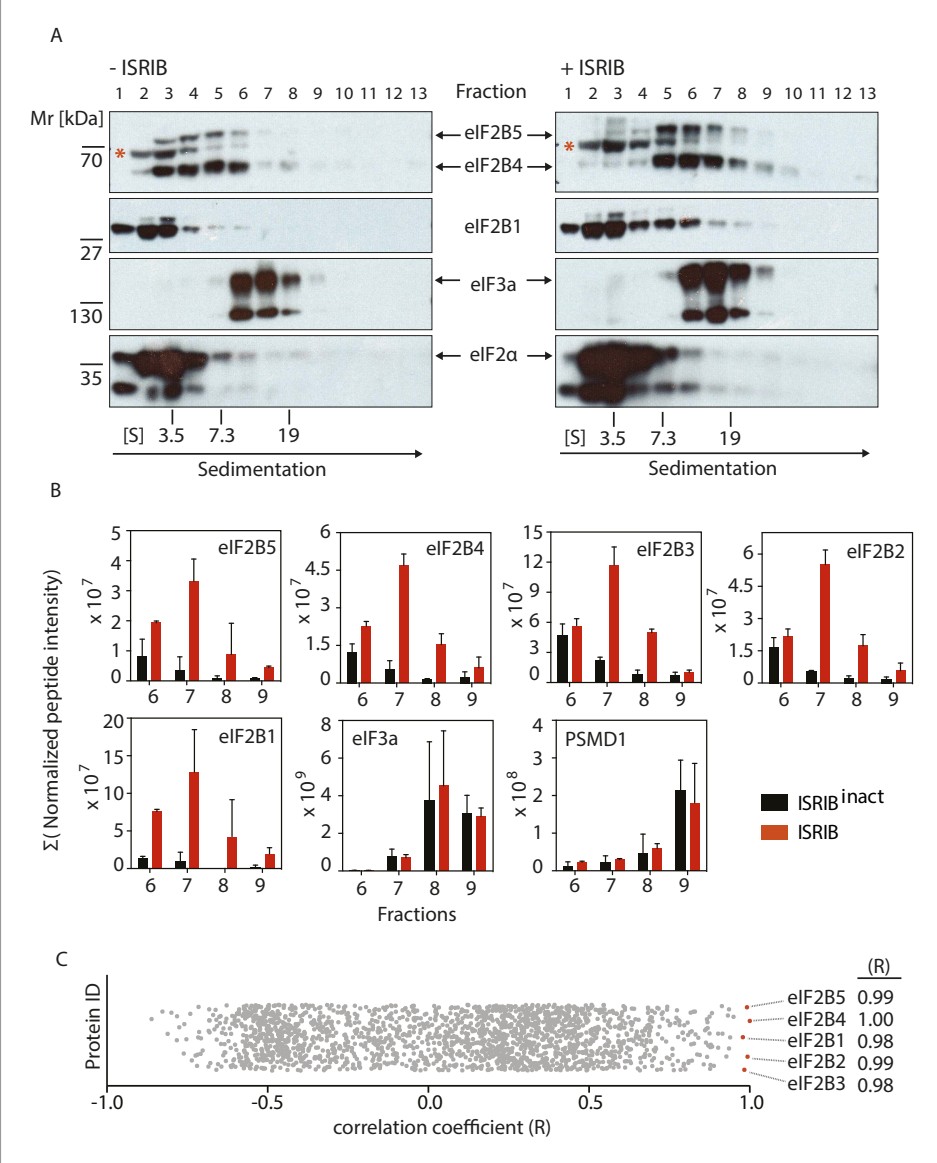

**Figure 3**. ISRIB induces dimerization of eIF2B in cells. (**A**) HEK293T cells were treated with or without 200 nM ISRIB and clarified lysates were loaded on a 5–20% sucrose gradient and subjected to centrifugation. 13 equal-size fractions were collected, protein was precipitated and run on a SDS-PAGE gel and immunoblotted with the indicated antibodies. The red asterisk indicates a background band that cross-reacts with the eIF2B4 antibody. Sedimentation was from left to right. Gradients were calibrated (in Svedberg units, 'S') with ovalbumin (S = 3.5; Mr = 44 kD); aldolase (S = 7.3; Mr = 158 kD) and thyroglobulin (S = 19; Mr = 669 kD). Shown is a representative blot (N = 3). (**B**) HEK293T cells and lysates were treated with 200 nM ISRIB or 200 nM ISRIB[inact] (ISRIB-A18; *Figure 3—figure supplement 1*) and clarified lysates were loaded on a 5–20% sucrose gradient and subjected to centrifugation. 13 equal sized fractions were collected and fractions 6–9 were precipitated, trypsinized and subjected to mass spectrometric analysis. The sum of the normalized peptide intensity of each eIF2B subunit as well as two control proteins, eIF3a and PSMD1 in each fraction was plotted. Two biological replicates were analyzed per condition (N = 2, ±SEM). The number of peptides and peptide intensity in fractions 6–9 for all proteins identified are listed in *Figure 3—source data 1*. (**C**) Correlation coefficient (R) of the sum of the normalized peptide intensity profile through fractions 6–9 for each protein identified in the analysis with respect to eIF2B4 was plotted. The Correlation coefficient (R) of the sum of the normalized peptide intensity profile with respect to eIF2B4 of each protein identified are listed in *Figure 3—source data 2*.

*Figure 3. continued on next page*

*Figure 3. Continued*

The following source data and figure supplements are available for figure 3:

**Source data 1**. Number of peptides and peptide intensity in fractions 6–9 for all proteins identified.

**Source data 2**. Correlation coefficient (R) of the sum of the normalized peptide intensity profile through fractions 6–9 with respect to eIF2B4 for each protein identified.

**Figure supplement 1**. Structures of ISRIB (ISRIB-A1) and ISRIB[inact] (ISRIB-A18).

**Figure supplement 2**. Analysis of the gradients subjected to mass spectrometric analysis in *Figure 3B*.

method relies on the principle that ligand binding can stabilize protein folding and hence increase the protein's resistance to heat denaturation. To this end, we incubated a cell lysate with and without ISRIB and then heated aliquots to different temperatures, followed by centrifugation to separate soluble from precipitated denatured proteins. We then analyzed the soluble fractions by Western blotting with antibodies against eIF2B1, eIF2B4 and eIF2B5. When the lysate was pre-incubated with ISRIB, we observed an increase in thermal stability of eIF2B4 (*Figure 4*, lanes 4 and 5, arrows). Although slight, the increase was highly reproducible and, as was the case for the analysis of the eIF2B shift in the sucrose gradients shown in *Figure 3*, a background band that cross-reacts with the anti-eIF2B4 antibody (red asterisk) provided a convenient internal control for the exclusive stabilization of eIF2B4. By contrast, no ISRIB-dependent increase in thermal stability was observed with the two other eIF2B subunits analyzed (eIF2B1 and eIF2B5), or with the translation initiation factors eIF2α or eIF3a (*Figure 4*). This analysis suggests that eIF2B subunits act autonomously in this assay, as eIF2B4 was stabilized while other subunits denatured and precipitated. We conclude that ISRIB binds eIF2B4 eliciting this stabilization.

## ISRIB enhances the GEF activity of eIF2B

To explore the functional consequences of ISRIB binding on eIF2B's GEF activity, we directly tested its effect on the rate of GDP release from eIF2. To this end, we pre-loaded purified eIF2 with radioactive

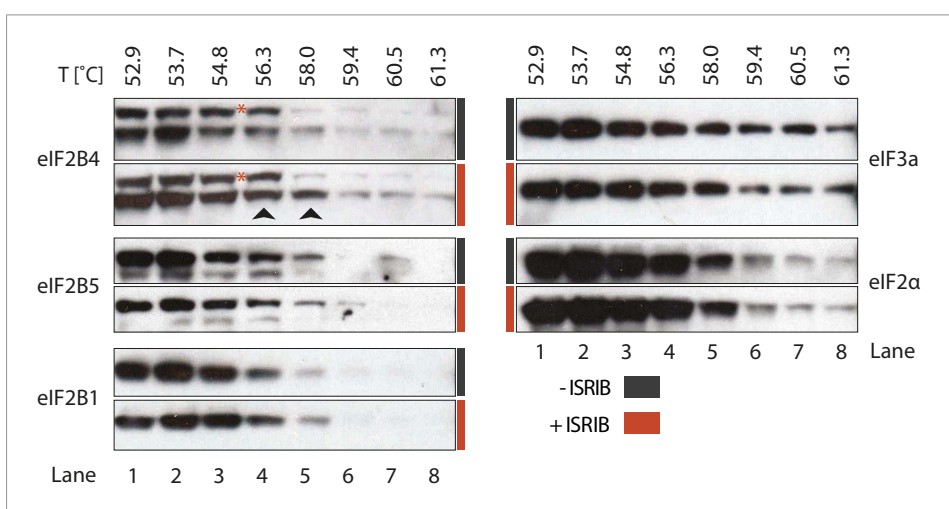

**Figure 4**. ISRIB enhances the thermo-stability of the regulatory subunit of eIF2B. Clarified HEK293 cell lysates were treated with DMSO (-ISRIB) or with 200 nM ISRIB (+ISRIB) for 20 min. Treated and untreated lysates were partitioned into smaller aliquots and heated to different temperatures for 3 min and then centrifuged to remove precipitated proteins. The supernatant fraction was loaded onto a SDS-PAGE gel and immunoblotted with the indicated antibodies. The red asterisk indicates a background band that cross-reacts with the eIF2B4 antibody. Shown is a representative blot (N = 3).

GDP ([³H]-GDP) and measured the fraction that remained bound as a function of time in the presence of an excess of unlabeled GDP. As expected, the intrinsic rate of nucleotide release was slow; after 20 min of incubation, only 20% of [³H]-GDP dissociated from the eIF2 complex (*Figure 5A*, black dashed line). The intrinsic rate of GDP release was not affected by the addition of ISRIB (*Figure 5A*, red dashed line). Upon addition of eIF2B, we observed a significant increase in the rate of GDP release ($t_{1/2}$ = 3.2 min), leading to an 80% release after 10 min (*Figure 5A*, solid black line). Excitingly, GDP release was threefold faster upon addition of ISRIB ($t_{1/2}$ = 1.1 min) (*Figure 5A*, solid red line).

We next tested the behavior of phosphorylated eIF2 (eIF2-P) in these assays. To this end, we generated eIF2-P by incubating eIF2 with recombinantly expressed PERK kinase and ATP. We next loaded eIF2-P with [³H]-GDP and measured GDP release. As expected from the known inhibitory role of eIF2α phosphorylation on eIF2B, GDP release from eIF2-P remained virtually unchanged in the presence of eIF2B (*Figure 5B*, black solid line). We next asked whether ISRIB allows eIF2-P to be a substrate for eIF2B. Our data show that ISRIB did not stimulate GDP release from eIF2-P (*Figure 5B*, red solid line), indicating that this is not the case. We next explored whether ISRIB can overcome the inhibitory effects

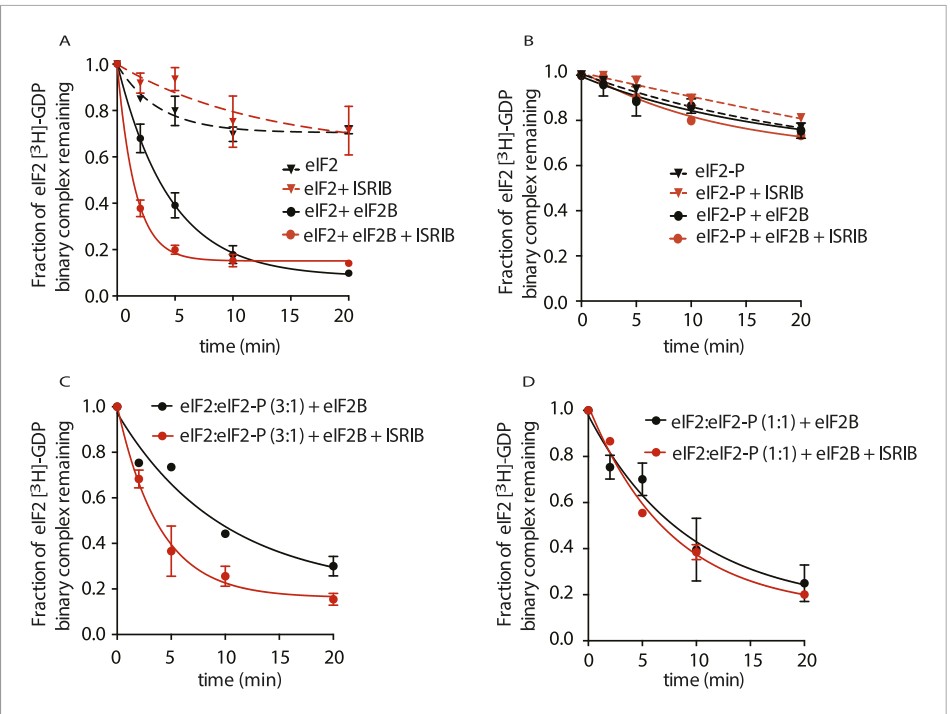

**Figure 5**. ISRIB enhances the GEF activity of eIF2B in vitro. eIF2 was preloaded with [³H]-GDP and the fraction of binary complex remaining was measured by filter binding. Partially purified eIF2B or buffer was added at t = 0 min. An aliquot of the reaction was stopped at the indicated times, filtered through a nitrocellulose membrane and radioactivity was measured. (**A**) Purified eIF2 was incubated with buffer (± 100 nM ISRIB, dashed lines) or partially purified eIF2B (± 100 nM ISRIB, solid lines) for the indicated times and the remaining fraction of [³H]-GDP-eIF2 was measured (N = 3, ± SD). (**B**) Purified and phosphorylated eIF2 (eIF2-P) was preloaded with [³H]-GDP and incubated with buffer (± 100 nM ISRIB, dashed lines) or partially purified eIF2B (± 100 nM ISRIB, solid lines) for the indicated times and the remaining fraction of [³H]-GDP-eIF2 was measured (N = 2, ± SD). (**C**) eIF2 was preloaded with [³H]-GDP and mixed with eIF2-P at a ratio of 3:1 and then incubated with eIF2B with or without 100 nM ISRIB for the indicated times and the remaining fraction of [³H]-GDP-eIF2 was measured (N = 2, ± SD). (**D**) eIF2 was preloaded with [³H]-GDP and mixed with eIF2-P at a ratio of 1:1 and then incubated with eIF2B with or without 100 nM ISRIB for the indicated times and the remaining fraction of [³H]-GDP-eIF2 was measured (N = 2, ± SD). Purified human eIF2 and partially purified rabbit reticulocyte eIF2B are shown in *Figure 5—figure supplement 1*.

The following figure supplement is available for figure 5:

**Figure supplement 1**. Purified human eIF2 (panel **A**, lane 2), recombinant GST-PERK (panel **A**, lane 1) and partially purified rabbit reticulocyte eIF2B (panel **B**) were analyzed by SDS-PAGE and stained with Coomassie blue dye.

of eIF2-P on eIF2B. To this end, we tested if ISRIB can promote GDP release from unphosphorylated eIF2 in the presence eIF2-P by mixing [$^3$H]-GDP-loaded eIF2 with eIF2-P in a 3:1 or 1:1 ratio. Although the exchange reaction was slower, ISRIB stimulated GDP release at the eIF2:eIF2-P ratio of 3:1 (-ISRIB: $t_{1/2}$ = 6.7 min, vs + ISRIB: $t_{1/2}$ = 2.7 min) (*Figure 5C*), whereas we observed hardly any stimulation at the 1:1 ratio (-ISRIB: $t_{1/2}$ = 6.4 min, vs + ISRIB: $t_{1/2}$ = 5.3 min) (*Figure 5D*). Thus, the relative ratio of substrate (eIF2) to inhibitor (eIF2-P) emerges as an important parameter affecting ISRIB's ability to modulate eIF2B activity. Taken together, these functional data underscore the notion that ISRIB acts as an activator of eIF2B and that ISRIB alleviates inhibition by eIF2-P, as long as eIF2-P is present below threshold levels.

## Discussion

The ISR is controlled by phosphorylation of the general eukaryotic translation initiation factor eIF2. Multiple cell signaling pathways converge at a single phosphorylation site on its α-subunit where phosphorylation of Ser-51 modulates eIF2α's interaction with its dedicated, multi-subunit guanine nucleotide-exchange factor (GEF) eIF2B. We previously identified and characterized a potent small molecule ISR inhibitor (ISRIB) with good pharmacological properties and showed that it renders cells insensitive to eIF2α phosphorylation upon ISR induction and enhances cognitive function in rodents (*Sidrauski et al., 2013*). Within a few minutes after administration, ISRIB reverses the effects triggered by eIF2α phosphorylation dissolving RNA stress granules and restoring translation of inhibited mRNAs while reversing de-repression of uORF-containing mRNAs (*Sidrauski et al., 2015*). Because ISRIB was identified in a phenotypic cell-based screen, its mechanism of action remained obscure. Here, we report the identification of eIF2B as the molecular target of ISRIB. To this end, we used reporter-based shRNA screening, structure–function analyses of ISRIB analogs, biochemical characterization of eIF2B oligomerization and thermal stability, and enzymatic analyses of eIF2B's GEF activity. The results of our multipronged approach provide a rationale for why ISRIB analogs exhibit twofold symmetry, showed ISRIB-mediated stabilization and activation of eIF2B dimers, and suggested eIF2B4, also known as its δ-subunit, as a candidate to contain the ISRIB binding site. In the course of this work, we also developed more active ISRIB analogs, improving potency by almost 10-fold and lowering $EC_{50}$ values into the high picomolar range in cell culture.

### How does ISRIB modulate eIF2B?

In this work, ISRIB emerged as an eIF2B activator. First, ISRIB promoted the formation of or stabilized eIF2B dimers ('[eIF2B]$_2$') and enhanced GEF activity in biochemical assays. Second, knockdown of both eIF2B4 and eIF2B5 subunits rendered cells resistant to the action of ISRIB, presumably because under these conditions the total amount of eIF2B that can be activated in cells is reduced. Note that the three other subunits of eIF2B were not represented in our focused shRNA library and therefore could not have been identified in the screen. Functioning as an activator, ISRIB joins the still sparsely populated group of unnatural small molecule enzyme activators, while the vast majority of synthetic small molecules that modulate enzyme activity are inhibitors (*Wiseman et al., 2010*; *Zorn and Wells, 2010*; *Wang et al., 2014*). Conversely, knockdown of eIF4G1 sensitized cells to ISRIB. This can be explained because, under conditions of reduced eIF4G1, overall cap-dependent translation initiation is reduced. A lower concentration of ISRIB could then suffice to generate sufficient amounts of GTP-loaded eIF2 to maintain normal rates of translation, even in the presence of eIF2α-P. Intriguingly, knockdown of other components of the cap-binding complex, such as eIF4A1, or components of the eIF3 complex, such as eIF3f and eIF3b, not only reduced sensitivity to ISRIB but also affected induction of the reporter upon ER stress alone. In agreement with studies in yeast and plants (*Szamecz et al., 2008*; *Roy et al., 2010*), knockdown of the eIF3 subunits in the library (eIF3a, eIF3b, and eIF3f) reduced translational induction of the reporter, presumably due to eIF3's stimulatory effects on re-initiation after translation of short uORFs. Our data therefore provide the first evidence that the mechanism of re-initiation may be similar in mammalian cells.

The differences observed between assorted initiation factors on reporter expression is likely to reflect the extent to which translation initiation was reduced under the different knockdown conditions. Importantly however, only knockdown of the eIF2B subunits targeted by shRNAs in the library conferred resistance to ISRIB.

We previously proposed two models that could explain how ISRIB renders cells resistant to the inhibitory effects of eIF2α-P (*Sidrauski et al., 2013*). First, ISRIB could weaken the effect of eIF2α-P on eIF2B by interfering with its tight and non-productive binding. In this way, more eIF2B would be available to reload eIF2 with GTP. Second, ISRIB could enhance the basal activity of eIF2B so that the fraction not engaged with eIF2α-P would produce sufficient levels of ternary complex to sustain

translation in cells. Currently, our in vitro enzymatic data do not allow us to distinguish between these models. While we showed that the rate of GDP release from purified eIF2 by eIF2B was significantly enhanced upon addition of ISRIB (and therefore can explain the effect of ISRIB in living cells), we do not know what fraction of our eIF2 preparation was isolated in a eIF2α(Ser-51)-phosphorylated state. ISRIB could thus either increase the GEF activity of eIF2B on eIF2 or diminish the inhibitory effect of a small amount eIF2-P present in the assay, akin to the regime that we directly tested by adding increasing amounts of in vitro phosphorylated eIF2 to the assay. Our analyses confirmed however that eIF2α-P is not a substrate for eIF2B (in agreement with previous reports [*Kimball et al., 1998*]), and determined that ISRIB does not enable eIF2B to use eIF2-P as a substrate.

While catalyzing guanine nucleotide exchange on other GTPases can be effected by relatively simple enzymes, eIF2B is a complex molecular machine composed of five different subunits. Much remains uncertain about the structural arrangement of the subunits and how eIF2B's activity is regulated (*Jennings and Pavitt, 2014*). Similarly, how ISRIB exerts its effects on eIF2B remains unknown. eIF2B subunits are organized into two modules, called the catalytic (eIF2B3 and eIF2B5) and regulatory (eIF2B1, eIF2B2 and eIF2B4) sub-complexes, containing two and three homologous proteins, respectively. The subunits of the regulatory subcomplex are characterized by highly homologous Rossman folds that bind nucleotides and are adorned by N-terminal extensions of lesser homology between the subunits. Intriguingly, recombinantly expressed eIF2B1 purified and crystallized as a stable homodimer, with an extensive buried interface contributed by the nucleotide-binding domains (*Bogorad et al., 2014*). The residues contributing to the interface are highly conserved among its homologs in the complex. Combined with the SAR analyses indicating ISRIB's obligate twofold symmetry, the discovery that $(eIF2B)_2$ exist in both yeast and mammalian cells was instrumental in suggesting to us that eIF2B is the target of ISRIB (*Gordiyenko et al., 2014*; *Wortham et al., 2014*). According to this model, ISRIB binds to two regulatory eIF2B subunits that form part of the interface linking two pentamers.

Native mass spectrometry of mammalian eIF2B revealed the existence of stable subcomplexes that lack the eIF2B1 subunit, indicating that this subunit is more loosely associated, as we confirmed here by sedimentation of the non-ISRIB treated control extracts (*Wortham et al., 2014*). We have shown by biochemical analysis that ISRIB binding stabilizes $(eIF2B)_2$, rendering it resistant to dissociation of eIF2B1 in the high-salt buffers used in the sucrose gradient analysis. Importantly, we showed by mass spectrometric proteomic analysis that no other protein co-profiled with $(eIF2B)_2$ in the gradients, demonstrating that the observed ISRIB-dependent effects were confined exclusively to eIF2B subunits.

Given the relative stability of the eIF2B1 homodimer ($K_d < 1$ nM; [*Bogorad et al., 2014*]) and our observation that ISRIB stabilized complete $(eIF2B)_2$, it is likely that two opposing eIF2B1 subunits form an essential part of the interface that links two eIF2B pentamers. ISRIB could favor this interaction by adding to the affinity provided by a $(eIF2B1)_2$ tether via the stabilization of an additional interface formed between homologous regions of two eIF2B4 subunits. This view would be in agreement with our data that showed protection by ISRIB of eIF2B4 to thermal denaturation. For symmetry reasons, as elegantly discussed in (*Bogorad et al., 2014*), this arrangement would leave the interfaces of the two identical eIF2B2 subunits in the complex unpaired. Alternatively, ISRIB may stabilize interfaces between eIF2B4 in one eIF2B pentamer and eIF2B2 in an opposing pentamer. If this were the case, ISRIB would bind at a pseudo-symmetric interface formed by two different, yet strongly homologous components. We note in this scenario, two ISRIB molecules binding to two identical interfaces of opposite polarity (eIF2B2 → eIF2B4 and eIF2B4 → eIF2B2) may bind and stabilize one $(eIF2B)_2$, which may contribute to its potency. This would open the possibility that design and synthesis of non-symmetric analogs could further improve ISRIB's efficacy. A definite assignment of ISRIB's binding site will have to await the structural determination of ISRIB-bound $(eIF2B)_2$ or genetic analyses in which loss-of-function mutations are suppressed by compensating changes in ISRIB analogs.

Consistent with the notion that the regulatory sub-complex provides binding sites for eIF2, mutations in eIF2B in yeast that render cells resistant to phosphorylation of eIF2α map to eIF2B1 and eIF2B4 (*Pavitt et al., 1997*). Moreover, two different variants in mammalian eIF2B4 (generated by alternative splicing) contain different N-terminal extension domains and exclusive expression of the longer variant desensitizes cells to eIF2α phosphorylation (*Martin et al., 2010*), phenocopying the effects elicited by ISRIB in mammalian cells. In the structure of $(eIF2B1)_2$ the N-terminal domains reach across the interface and interact with the nucleotide binding domain of the partnering eIF2B1 molecule. We speculate that the extended N-terminal domain of eIF2B4 may stabilize $(eIF2B)_2$, mimicking the effects of ISRIB.

## Importance of eIF2-mediated translational control in disease

Phosphorylation of eIF2 is important in long-term depression (LTD), and we have recently shown that this modulation of synaptic plasticity can explain cognitive enhancement elicited by ISRIB treatment of wild type rodents (*Di Prisco et al., 2014*). Engagement of metabotropic glutamate receptors (mGluR) in post-synaptic hippocampal cells leads to eIF2 phosphorylation and preferential translation of neuronally expressed oligophrenin-1 (encoded by OPHN1), a protein that mediates the initial steps of downregulation of postsynaptic AMPA receptors by endocytosis (*Nadif Kasri et al., 2011*). Like ATF4, the 5′-UTR of OPHN1 mRNA contains two uORFs that repress expression of the downstream coding sequence unless eIF2 is phosphorylated. Importantly, both genetic ablation of eIF2 phosphorylation and treatment with ISRIB but not the inactive analog ISRIB-A18 abolished the reduction in surface AMPARs and blocked mGluR-LTD (*Di Prisco et al., 2014*). These findings hold promise that targeting the effects of phosphorylation of eIF2 by pharmacologically modulating eIF2B with drugs such as ISRIB could result in therapies for cognitive disorders. Activation of the ISR with its characteristic increase in eIF2 phosphorylation has been reported in numerous neurodegenerative diseases, including Alzheimer's disease, Parkinson's disease, Frontotemporal Dementia, Amyotrophic Lateral Sclerosis, and prion neurodegenerative diseases, but its role in disease progression has just recently begun to be interrogated (*Kim et al., 2013*; *Leitman et al., 2014*; *Ma et al., 2013*; *Moreno et al., 2013*, *2012*).

The importance of eIF2 and eIF2B in brain function is underscored by the existence of mutations in these factors that cause human disease. A familial intellectual disability syndrome was mapped to a mutation in the γ subunit of eIF2 (encoded by EIF2S3). When an analogous mutation was introduced into yeast cells, it impaired eIF2-mediated translation initiation (*Borck et al., 2012*). Mutations in the different subunits of eIF2B cause childhood ataxia with central nervous system (CNS) hypomyelination (CACH) or vanishing white matter disease (VWMD). All affected individuals have two altered copies of a single eIF2B gene (autosomal recessive inheritance) and the majority are missense mutations that cause a single amino acid change while the remainder is a mixture of premature nonsense mutations, some causing a frame-shift and others altered splicing. All subunits of eIF2B are essential and the biochemical analysis of 40 different VWMD mutations revealed that the majority are hypomorphs, that is, cause partial loss-of function of eIF2B GEF activity (*Leegwater et al., 2001*; *Li et al., 2004*; *Fogli and Boespflug-Tanguy, 2006*). Whether ISRIB can reverse the deleterious effects of mutations in eIF2B in VWMD patients is not known, but we speculate that it may protect from a further reduction in GEF activity by stress-induced eIF2α-P. Intriguingly, the onset of VWMD is varied but generally exacerbated by head trauma and febrile illnesses. Interestingly, two VWMD mutations have been characterized that affect the integrity and dimerization of the eIF2B complex. A mutation in eIF2B1(V183F) maps to the dimerization interface and the mutant recombinant protein is predominantly in the monomeric form and a mutation in eIF2B4(A391D) affects complex integrity in the absence of eIF2B1 and dimerization (*Wortham et al., 2014*). ISRIB induces dimerization and complex stability and thus may rescue the effects of such mutations.

Given the wide spectrum of potential applications for ISRIB in neurological diseases, the identification of its molecular target is an important step. Having established a proof-of-principle that eIF2B can be pharmacologically modulated, now enables directed screening efforts to identify new series of compounds and thereby enhance the probability of developing clinically useful pharmaceuticals that address currently unmet needs.

## Note added at proof

While this work was under review, Sekine et al. reported the independent identification of eIF2B as the molecular target of ISRIB (*Sekine et al., 2015*).

# Materials and methods

## Chemicals

Thapsigargin (Tg) was obtained from Sigma–Aldrich (St Louis, MO). Tunicamycin (Tm) was obtained from Calbiochem EMB Bioscience (Billerica, CA). The GSK PERK inhibitor (G797800) was obtained from Toronto Research Chemicals (North York, ON, Canada).

## Cell culture

HEK293T and K562 cells were maintained at 37C, 5% $CO_2$ in either DMEM (HEK293T) or RPMI (K562) media supplemented with 10% FBS, L-glutamine and antibiotics (penicillin and streptomycin).

## shRNA screening reporter cell line

The lentiviral reporter vector, pMK1163, contains a CMV promoter driving expression of a fusion transcript with the following elements: the 5′ end of the human ATF4 mRNA up to the start codon of the ATF4-encoding ORF, an ORF encoding Venus (adapting a previously published strategy [*Lu et al., 2004*; *Vattem and Wek, 2004*]), followed by an IRES driving translation of tagBFP. The elements of this vector were generated as follows: we PCR-amplified the ATF4 region from human cDNA prepared from K562 cells using primers oMK305 (5′-CGTACTCGAGTTTCTACTTTGCCCGCCCA CAG-3′) and oMK306 (5′-GCTCCTCGCCCTTGCTCACCATGTTGCGGTGCTTTGCTGGAATCG-3′). Venus was amplified from DAA307 (gift from Diego Acosta-Alvear), using primers oMK272 (5′-ATGGTGAGCAAGGGCGAGGAGC-3′) and oMK308 (5′-GCTAGAATTCTTACTTGTACAGCTCGTC CATGCC-3′). The ATF4-Venus fusion was generated by PCR reaction using the two PCR products described above as templates, and oMK305 and oMK308 as primers. The EMCV IRES was amplified from plasmid pPPCX-IRES-GFP (gift from Diego Acosta-Alvear). tagBFP was amplified from a tagBFP plasmid (Evrogen, Moscow, Russia). The plasmid pMK1163 is in the lentiviral vector pSicoR (*Ventura et al., 2004*), and its sequence is provided in *Figure 1—source data 1*. Human K562 cells were transduced with pMK1163 and monoclonal cell lines were generated using FACS. One clone was selected as our reporter cell line based on low base-line expression of Venus and high expression following thapsigargin treatment (high dynamic range).

## Pooled shRNA screen

The reporter cell line was transduced with a pooled next-generation shRNA library. We used a sub-library that targets 2933 human genes associated with proteostasis, each with on average 25 independent shRNAs, and contains >1000 negative control shRNAs. After transduction, transduced cells were selected with puromycin (0.65 µg/ml) for 2 days, and then grown in the absence of puromycin for 2 days. Cells were then separated into two populations, which were treated for 7 hr with either 300 nM thapsigargin alone or 300 nM thapsigargin and 15 nM ISRIB. Cells were then sorted based on reporter fluorescence using a BD FACS Aria2. Cells from the thirds of the population with the highest and lowest reporter levels were collected. Genomic DNA was isolated from FACS-sorted populations, and shRNA-encoding cassettes were PCR-amplified and subjected to deep sequencing as previously described (*Kampmann et al., 2014*). Using our previously described analysis pipeline (*Kampmann et al., 2013*, *2014*), we calculated a quantitative phenotype $\varepsilon$ for each shRNA, which represents the $\log_2$ ratio of its frequency in the high-fluorescence population over its frequency in the low-fluorescence population, from which the median of the negative control phenotypes was subtracted (*Kampmann et al., 2013*). For each gene, $\varepsilon$ phenotypes for the ~25 shRNAs targeting the gene were compared to $\varepsilon$ phenotypes for the negative control shRNAs, and p values were calculated using the Mann–Whitney U test to detect genes whose knockdown significantly modulated activation of the uORFs-ATF4-venus reporter in response to thapsigargin in the absence or presence of ISRIB. p values for all 2933 genes targeted by the sublibrary we used are listed in *Figure 1—source data 2*.

## Cell-based assay to measure the potency of ISRIB analogs

HEK293T cells carrying an ATF4 luciferase reporter (as previously described in [*Sidrauski et al., 2013*]) were plated on poly-lysine coated 96 well plates (Greiner Bio-One, Monroe, NC) at 30,000 cells per well. Cells were treated the next day with tunicamycin (1 µg/ml) and different concentrations (serial dilution) of each compound for 7 hr. Luminescence was measured using One Glo (Promega, Madison, WI) as specified by the manufacturer. $EC_{50}$ values were calculated by plotting $\log_{10}$ [µM] for each compound as a function of the relative luminescence intensity or response. The $EC_{50}$ corresponds to the concentration that provokes a half-maximal response.

## Sucrose gradients

HEK293T cells were plated on 150 mm plates, treated with or without 200 nM ISRIB for 20 min, washed twice with ice-cold PBS, collected and centrifuged for 3 min at 800 rcf at 4°C. The pellets were

resuspended in ice-cold lysis buffer: 50 mM Tris pH = 7.5, 400 mM KCl, 4 mM Mg(OAc)$_2$, 0.5% Triton X-100 and protease inhibitors (EDTA-free protease inhibitor tablets, Roche, South San Francisco, CA). The lysates were clarified at 20,000×$g$ for 15 min at 4˚C and the supernatant was then subjected to a high-speed spin at 100,000×$g$ for 30 min at 4˚C to pellet the ribosomes. The supernatants were then loaded on a 5–20% sucrose gradient and centrifuged in a SW55 rotor for 14 hr at 40,000 rpm 4˚C. 13 fractions were collected, protein was chloroform-methanol precipitated, resuspended in SDS-PAGE loading buffer and loaded on SDS-PAGE 10% gels (Bio-Rad, Hercules, CA).

### Protein analysis

Proteins were transferred to nitrocellulose and probed with primary antibodies diluted in phosphate-buffered saline supplemented with 0.1% Tween 20 and 5% bovine serum albumin. The following antibodies were used: eIF2B1 (1:1000; Proteintech 18010-1-AP, Chicago, IL), eIF2B2 (1:500; Proteintech 11034-1-AP), eIF2B4 (1:1000; Proteintech 11332-1-AP), eIF2B5 (1:500; Santa Cruz Biotechnologies sc-5558, Dallas, TX), eIF3a (1:1500; Cell Signaling Technology #3411, Danvers, MA) and eIF2α (1: 1500; Cell Signaling Technology #5324). Following primary antibody incubation, either HRP-conjugated secondary antibody (Promega) or IRdye conjugated secondary antibodies (LI-COR Biosciences, Lincoln, NE) was used. Immunoreactive bands were detected using either enhanced chemi-luminescence (Bio-Rad) or the LI-COR Odyssey imaging system.

### Mass spectrometry of sucrose gradient fractions

HEK293T cells were treated with ISRIB or ISRIB[inact] (ISRIB-A18, *Figure 3—figure supplement 1*) at 200 nM for 20 min. Cells were then subjected to three liquid nitrogen freeze–thaw cycles in a modified lysis buffer devoid of Triton X-100 and supplemented with ISRIB or ISRIB[inact] at 50 nM. Lysates were loaded onto a 5–20% sucrose gradient. Proteins in fractions 6–9 were chloroform-methanol precipitated and re-suspended in 0.1 M tetraethylammonium bromide (TEAB), 150 mM NaCl and 8M Urea and digested with trypsin as previously described (*Ramage et al., 2015*).

Digested peptide mixtures were analyzed in technical duplicate by LC-MS/MS on a Thermo Scientific LTQ Orbitrap Elite mass spectrometry system equipped with a Proxeon Easy nLC 1000 ultra high-pressure liquid chromatography and autosampler system. Samples were injected onto a C18 column (25 cm × 75 µm I.D.) packed with ReproSil Pur C18 AQ (1.9 µm particles) in 0.1% formic acid and then separated with a 1-hr gradient from 5% to 30% ACN in 0.1% formic acid at a flow rate of 300 nl/min. The mass spectrometer collected data in a data-dependent fashion, collecting one full scan in the Orbitrap at 120,000 resolution followed by 20 collision-induced dissociation MS/MS scans in the dual linear ion trap for the 20 most intense peaks from the full scan. Dynamic exclusion was enabled for 30 s with a repeat count of one. Charge state screening was employed to reject analysis of singly charged species or species for which a charge could not be assigned.

Raw mass spectrometry data were analyzed using the MaxQuant software package (version 1.3.0.5) (*Cox and Mann, 2008*). Data were matched to the SwissProt human proteins (downloaded from UniProt on 2/15/13, 20,259 protein sequence entries). MaxQuant was configured to generate and search against a reverse sequence database for false discovery rate calculations. Variable modifications were allowed for methionine oxidation and protein N-terminus acetylation. A fixed modification was indicated for cysteine carbamidomethylation. Full trypsin specificity was required. The first search was performed with a mass accuracy of ± 20 parts per million and the main search was performed with a mass accuracy of ± 6 parts per million. A maximum of five modifications were allowed per peptide. A maximum of two missed cleavages were allowed. The maximum charge allowed was 7+. Individual peptide mass tolerances were allowed. For MS/MS matching, a mass tolerance of 0.5 Da was allowed and the top six peaks per 100 Da were analyzed. MS/MS matching was allowed for higher charge states, water and ammonia loss events. The data were filtered to obtain a peptide, protein, and site-level false discovery rate of 0.01. The minimum peptide length was 7 amino acids. Results were matched between runs with a time window of 2 min for technical duplicates.

### CETSA

CETSA were adapted from a previously described protocol (*Martinez Molina et al., 2013*). HEK293T cells were lysed in a buffer containing: 50 mM Tris pH = 7.5, 400 mM KCl, 4 mM Mg(OAc)$_2$, 0.5% Triton X-100 and protease inhibitors (EDTA-free protease inhibitor tablets, Roche). The lysates

were clarified at 20,000×g for 15 min at 4°C. The supernatant was then incubated with ISRIB (1 µM, 0.1% DMSO) or DMSO (0.1%) at 30°C for 20 min, and subsequently spun at 100,000×g for 30 min at 4°C to pellet ribosomes. Supernatants following the high-speed spin were divided into PCR tubes and subjected to a gradient of temperatures for 3 min using the thermal cycler's built-in gradient function, such that column one corresponded to 52°C and column 12 corresponded to 62°C (Tetrad 2 Thermal Cycler, Bio-Rad). Samples were allowed to cool for 3 min at room temperature, transferred to microfuge tubes, and spun at 20,000×g for 20 min at 4°C to separate the soluble fraction from the insoluble precipitates. The soluble fraction was then loaded on a 10% SDS-PAGE gel (Bio-Rad) and analyzed by Western blotting as described above.

## Purification of eIF2B

Rabbit reticulocyte lysate was obtained from Greenhectares (http://greenhectares.com). eIF2B was purified as previously described (*Oldfield and Proud, 1992*). In brief, the reticulocyte lysate was thawed and protease inhibitor added (EDTA-free protease inhibitor tablets, Roche). Ribosomes were precipitated by centrifugation (45,000 rpm for 4.5 hr, Beckman 50.2 Ti at 4°C) and the supernatant was used as a source of eIF2B. KCl was added slowly to 100 mM final concentration and filtered using a 0.2 µM conical tube filter unit. The filtrate was loaded on a SP-Sepharose fast flow column (20 ml) pre-equilibrated with Buffer A (20 mM Hepes/NaOH pH = 7.6, 10% glycerol, 100 mM KCl, 0.1 mM EDTA and 2 mM DTT). A step gradient was used (100, 200 and 400 mM KCl). eIF2B eluted at 400 mM KCl. The eluate was diluted slowly by adding Buffer A (with no KCl) to 100 mM KCl and then loaded on a Q-Sepharose (20 ml) pre-equilibrated with Buffer A. A step gradient was used (300 mM and 500 mM KCl) with eIF2B eluting at 500 mM KCl. The eluate was dialyzed overnight with Buffer A and loaded to a Mono Q (5-50 GL, GE Healthcare, Wauwatosa, WI) equilibrated with buffer A (a continuous gradient 100–500 mM KCl was used) and eIF2B eluted at 350 mM KCl. The eluate was buffer exchanged with Buffer A and aliquots were flash frozen in liquid $N_2$.

## Purification of eIF2

Human eIF2 was purified from HeLa cells as described previously (*Fraser et al., 2007*). In brief, from the 40–50% ammonium sulfate precipitate of post-nuclear HeLa cell lysate, eIF2 was purified through a series of chromatographic steps which included a Mono Q 10/10 column (GE Healthcare), a Mono S 10/10 column (GE Healthcare), a CHT5-1 ceramic hydroxyapatite column (Bio-Rad), and a Superose 6 16/60 column (GE Healthcare). The protein was stored at −80°C in buffer containing 20 mM Hepes-K pH 7.5, 150 mM KCl, 1 mM DTT, and 10% glycerol.

## GDP dissociation assay

GDP dissociation assays were adapted from a previously described protocol (*Sokabe et al., 2012*). For each reaction purified eIF2 (21 pmol) was incubated with 0.6 µCi [³H]-GDP (40 Ci/mmol, PerkinElmer, Waltham, MA) in a reaction buffer (20 mM HEPES pH 7.5, 80 mM KCl, 1 mM DTT, 1 mg/ml creatine phosphokinase (EMD Millipore, Billerica, MA), 5% glycerol) without magnesium at 37°C for 10 min, and then further incubated with 1 mM Mg(OAc)$_2$ at 30°C for 3 min with or without ISRIB (100 nM) in a total volume of 60 µl. The reaction was initiatied by the addition of 60 nmol unlabeled GDP with or without eIF2B (0.6 µl of partially purified rabbit reticulocyte eIF2B, which correspond to approximately 0.3 pmoles of the complex). At each time point, an aliquot was taken (10 µl) and the reaction was stopped by addition to 300 µl ice-cold stop buffer (reaction buffer with 5 mM Mg(OAc)$_2$), immediately filtered through a HAWP nitrocellulose membrane filter (EMD Millipore) on a vacuum manifold, and washed twice with 1 ml ice-cold stop buffer. Filters were dried and remaining [³H]-GDP bound to eIF2 was counted by liquid scintillation in Ecoscint (National Diagnostics, Atlanta, GA). Data collected were fitted to a first-order exponential decay.

eIF2-P was synthesized by incubating eIF2 (1.76 µM) with recombinant GST-PERK (500 nM) at 37°C for 45 min in a reaction buffer containing: 0.5 mM ATP, 50 mM Tris–HCl pH 7.5, 4 mM MgCl$_2$, 100 mM NaCl, 1 mM *tris*(2-carboxyethyl)phosphine (TCEP), 1% glycerol. The phosphorylation reaction was stopped by the addition of 1 µM GSK PERK inhibitor (Toronto Research Chemicals) and 4 mM EDTA to chelate magnesium ions. For eIF2-P•GDP dissociation reactions (*Figure 5B*), eIF2-P (21 pmol) was loaded with [³H]-GDP. For experiments where eIF2 was mixed with eIF2-P (*Figure 5C,D*),

unphosphorylated eIF2 was loaded with [$^3$H]-GDP and mixed (3:1 or 1:1) with eIF2-P, which was not loaded with [$^3$H]-GDP, such that the sum of eIF2 and eIF2-P equaled 21 pmol. GDP dissociation assays were conducted as described above in the presence of 50 nM GSK PERK inhibitor to ensure that the residual PERK kinase did not phosphorylate eIF2 during the course of the dissociation assay.

## Purification of GST-PERK

Cytosolic human PERK was codon-optimized for *Escherichia coli* expression by Genewiz Inc. A construct was then cloned into a PGEX-6P-2 vector for expression using two rounds of In-Fusion cloning (Clontech, Mountain View, CA) (535–1093 Δ660–868). The cytosolic portion of PERK, lacking the unstructured loop region (amino acids 535–1093 Δ660–868) was then co-expressed with a tag-less lambda phosphatase to produce a fully dephosphorylated PERK protein in BL21 star (DE3) (Life Technologies, Carlsbad, CA). Cells were grown to an $OD_{600}$ of 0.5 before induction with 0.1 mM IPTG at 15°C for 24 hr. Cells where harvested and lysed using AVESTIN Emulsiflex-C3 in a buffer containing 50 mM Tris-HCl, pH 8.0, 500 mM NaCl, 5% glycerol, 5 mM TCEP (buffer A) and EDTA-free COMPlete protease inhibitor cocktail (Roche). The lysate was cleared by centrifugation at 100,000×g before batch-binding to a GST-Sepharose resin. The resin was washed 5 times with buffer A. The protein was loaded onto a HiTrap Q HP column to remove remaining lambda phosphatase. The PERK (535–1093 Δ660–868) protein was then concentrated and fractionated on a Superdex 200 GL (GE Healthcare) to remove protein aggregates.

## Chemical syntheses

### General methods

Commercially available reagents and solvents were used as received. Compounds **ISRIB-A1** and **ISRIB-A2** were prepared as previously reported (Sidrauski et al., 2013b). Compound **ISRIB-A7** was available commercially from Specs (The Netherlands). $^1$H NMR spectra were recorded on a Varian INOVA-400 400 MHz spectrometer and a Bruker Avance 300 300 MHz spectrometer. Chemical shifts are reported in δ units (ppm) relative to residual solvent peak. Coupling constants (*J*) are reported in hertz (Hz). LC-MS analyses were carried out using Waters 2795 separations module equipped with Waters 2996 photodiode array detector, Waters 2424 ELS detector, Waters micromass ZQ single quadropole mass detector, and an XBridge C18 column (5 μm, 4.6 × 50 mm). Microwave reactions were carried out in a CEM Discover microwave reactor.

## General procedure A for amide coupling

To a solution of the carboxylic acid (1 equiv.) in N,N-dimethylformamide, were sequentially added 1-hydroxybenzotriazole hydrate (1.2 equiv.), 1-(3-dimethylaminopropyl)-3-ethylcarbodiimide hydrochloride (1.2 equiv.), 2-(4-chlorophenoxy)-N-[(1r,4r)-4-aminocyclohexyl]acetamide trifluoroacetic acid (1.0 equiv., prepared as described in the synthesis of **ISRIB-A8**, below) and N,N-diisopropylethylamine (1.5 equiv). The reaction mixture was stirred at room temperature until judged complete by LC-MS and then diluted with water (2 ml). The mixture was vigorously vortexed, centrifuged and the water was decanted. This washing protocol was repeated with water (2 ml) and then with diethyl ether (2 ml). The wet solid was dissolved in dichloromethane (10 ml) and dried over anhydrous magnesium sulfate. The solids were removed by filtration and the filtrate was concentrated by rotary evaporation to obtain the product.

## General procedure B for amide coupling

To a solution of the carboxylic acid (2 equiv.) in N,N-dimethylformamide were sequentially added 1-hydroxybenzotriazole hydrate (2 equiv.), 1-(3-dimethylaminopropyl)-3-ethylcarbodiimide hydrochloride (2 equiv.), the diamine (1.0 equiv.) and N,N-diisopropylethylamine (6 equiv). The reaction mixture was stirred at room temperature until judged complete by LC-MS and then diluted with water. The precipitate formed was washed with water and 10% diethyl ether in dichloromethane. The precipitate was dried in vacuo to obtain the product.

## General procedure C for amide coupling

To a solution of (1r,4r)-cyclohexane-1,4-diamine (1 equiv.) in N,N-dimethylformamide were added the carboxylic acid (2 equiv.), 1-[bis(dimethylamino)methylene]-1H-1,2,3-triazolo[4,5-b]pyridinium 3-oxid

hexafluorophosphate (2.1 equiv.) and N,N-diisopropylethylamine (4 equiv.). The reaction mixture was vigorously stirred at room temperature until judged complete by LC-MS. Water (2 ml) was added. The mixture was centrifuged and the water was decanted. This washing protocol was repeated thrice and the resulting wet solid was concentrated down with toluene (10 ml) in a rotary evaporator. The residual product was washed with diethyl ether (10 ml) and concentrated using rotary evaporation to obtain the product.

## 2-(4-Chlorophenoxy)-N-{4-[2-(4-chlorophenoxy)acetamido]butyl} acetamide (ISRIB-A3)

To a solution of 1,4-diaminobutane (0.032 g, 0.2 mmol) in tetrahydrofuran (1.0 ml), were added 4-chlorophenoxyacetyl chloride (0.062 ml, 0.4 mmol) and N,N-diisopropylethylamine (0.173 ml, 1.0 mmol). The reaction mixture was stirred at room temperature for 20 hr and then partitioned between 1:1 mixture of water/dichloromethane (20 ml). The organic layer was washed with 10% aqueous potassium hydrogen sulfate, water and brine. The organic phase was then dried over magnesium sulfate, filtered, and concentrated to obtain a brownish orange solid. The brownish orange solid was triturated with diethyl ether and the resulting solids were separated by centrifugation and dried to obtain 26 mg (31%) of the title compound as tan powder. [1]H NMR (400 MHz, DMSO-d6) δ 8.06 (t, J = 5.6 Hz, 2H), 7.30–7.32 m, 4H), 6.93–6.95 (m, 4H), 4.43 (s, 4H), 3.08 (d, J = 5.7Hz, 4H), 1.37 (br. s, 4H) LC-MS: m/z = 425 [M + H, 35Cl ]+, 427 [M + H, 37Cl]+.

## 2-(4-Chlorophenoxy)-N-[(1r,3r)-3-[2-(4-chlorophenoxy)acetamido] cyclobutyl]acetamide (ISRIB-A4)

To a cooled (0°C) solution of tert-butyl N-[(1r,3r)-3-aminocyclobutyl]carbamate (0.05 g, 0.277 mmol) in 1,2-dichloroethane (1.38 ml), was added trifluoroacetic acid (1.38 ml). The reaction mixture was stirred at room temperature for 2 hr and then concentrated down to dryness to obtain 100 mg of (1r,3r)-cyclobutane-1,3-bis(aminium) ditrifluoroacetate which was used without further purification.

To a solution 4-chlorophenoxyacetic acid (0.19 g, 0.63 mmol) in N,N-dimethylformamide (1.0 ml) were sequentially added 1-hydroxybenzotriazole hydrate (0.12 g, 0.63 mmol), 1-(3-dimethylaminopropyl)-3-ethylcarbodiimide hydrochloride (0.175 g, 0.63 mmol), (1r,3r)-cyclobutane-1,3-bis(aminium) ditrifluoroacetate (0.1 g, 0.31 mmol) and N,N-diisopropylethylamine (0.34 ml, 1.91 mmol). The reaction mixture was stirred at room temperature for 2 hr and then subjected to conditions described in procedure B to afford 72 mg (54%) of the title compound. [1]H NMR (300 MHz, CDCl₃) δ 7.29–7.35 (m, 4H), 6.91 (dd, J = 9, 2.2 Hz, 4H), 6.80 (d, J = 7.6 Hz, 2H), 4.60–4.62 (m, 2H), 4.48 (s, 4H), 2.46–2.51 (m, 4H) LC-MS: m/z = 423 [M + H]+.

## 2-(4-Chlorophenoxy)-N-[(1s,3s)-3-[2-(4-chlorophenoxy)acetamido]cyclobutyl]acetamide (ISRIB-A5)

To a cooled (0°C) solution of tert-butyl N-[(1 s,3 s)-3-aminocyclobutyl]carbamate (0.05 g, 0.277 mmol) in 1,2-dichloroethane (1.38 ml), was added trifluoroacetic acid (1.38 ml). The reaction mixture was stirred at room temperature for 1.5 hr and then concentrated down to dryness to obtain 100 mg of (1 s,3 s)-cyclobutane-1,3-bis(aminium) ditrifluoroacetate which was used without further purification.

To a solution 4-chlorophenoxyacetic acid (0.19 g, 0.63 mmol) in N,N-dimethylformamide (1.0 ml) were sequentially added 1-hydroxybenzotriazole hydrate (0.12 g, 0.63 mmol), 1-(3-dimethylaminopropyl)-3-ethylcarbodiimide hydrochloride (0.175 g, 0.63 mmol), (1 s,3 s)-cyclobutane-1,3-bis(aminium) ditrifluoroacetate (0.1 g, 0.31 mmol) and N,N-diisopropylethylamine (0.34 ml, 1.91 mmol). The reaction mixture was stirred at room temperature for 2 hr. The reaction mixture was then diluted with 5% methanol in dichloromethane, washed with water and brine. The organic layer was dried over magnesium sulfate, filtered and concentrated. The crude mixture was purified by flash column chromatography (40% acetone/hexanes) to obtain 34 mg (25%) of the title compound. $^1$H NMR (300 MHz, CDCl$_3$) δ 7.26–7.29 (m, 4H), 6.84–6.87 (m, 4H), 6.77 (d, J = 6.5 Hz, 2H), 4.42 (m, 4H), 4.17–4.25 (s, 2H), 2.84–2.93 (m, 2H), 2.02–2.12 (m, 2H) LC-MS: $m/z$ = 423 [M + H]$^+$.

## 2-(4-Chlorophenoxy)-N-{3-[2-(4-chlorophenoxy)acetamido]propyl}acetamide (ISRIB-A6)

To a solution of 1,3-diaminopropane (0.017 ml, 0.2 mmol) in tetrahydrofuran (0.6 ml), was added 4-chlorophenoxyacetyl chloride (0.062 ml, 0.4 mmol) and N,N-diisopropylethylamine (0.08 ml, 0.5 mmol). The reaction mixture was stirred at room temperature for an hour and then partitioned between 1:1 mixture of water/dichloromethane (20 ml). The organic layer was washed with 10% aqueous potassium hydrogen sulfate, water and brine. The organic phase was then dried over magnesium sulfate, filtered and concentrated to obtain a brownish orange oil. The brownish orange oil was purified by flash column chromatography (5–80% acetone/dichloromethane) to obtain 41 mg (49%) of the title compound. $^1$H NMR (400 MHz, CDCl$_3$) δ 7.24–7.26( m, 4H), 7.15 (br.s, 2H), 6.85–6.87 (m, 4H), 4.45 (s, 4H), 3.08 (quint, J = 6.3 Hz, 4H), 1.37 (quint, J = 6.2 Hz, 2H) LC-MS: m/z = 411 [M + H, 35Cl ]+, 413 [M + H, 37Cl]+.

## 2-(4-Fluorophenoxy)-N-[(1r,4r)-4-[2-(4-chlorophenoxy)acetamido]cyclohexyl]acetamide (ISRIB-A8)

Step 1: To a mixture of *tert*-butyl N-[(1r,4r)-4-aminocyclohexyl]carbamate (0.750 g, 3.5 mmol) in THF (20 ml) were sequentially added N,N-diisopropylethylamine (0.914 ml, 5.25 mmol) and 4-chlorophenoxyacetyl chloride (0.573 ml, 3.78 mmol). The reaction mixture was vigorously stirred at room temperature for 3 hr and then diluted with water (100 ml). The precipitate was filtered and the solid was washed with water. The resulting solid was then diluted with diethyl ether and vacuum filtered. The filter cake was washed with diethyl ether. The residual ether was removed under vacuum to afford 1.103 g (82%) of *tert*-butyl N-[(1r,4r)-4-[2-(4-chlorophenoxy)acetamido]cyclohexyl]carbamate as a white solid. $^1$H NMR (400 MHz, DMSO-$d_6$) δ 7.88 (d, $J$ = 7.87 Hz, 1H), 7.25–7.37 (m, 2H), 6.93 (d, $J$ = 8.97 Hz, 2H), 6.68 (d, $J$ = 7.69 Hz, 1H), 4.41 (s, 2H), 3.51 (m, 1H), 3.13 (br. s., 1H), 1.72 (t, $J$ = 13.19 Hz, 4H), 1.34 (s, 9H), 1.09–1.30 (m, 4H); LC-MS: $m/z$ = 405 [M + Na, $^{35}$Cl ]$^+$, 407 [M + Na, $^{37}$Cl ]$^+$, 765 [2M + H, $^{35}$Cl × 2]$^+$, 767 [2M + H, $^{35}$Cl, $^{37}$Cl]$^+$.

Step 2: To a suspension of tert-butyl N-[(1r,4r)-4-[2-(4-chlorophenoxy)acetamido]cyclohexyl] carbamate (0.5 g, 1.31 mmol) in dichloromethane (9 ml) were sequentially added triethylsilane (0.3 ml, 1.88 mmol), water (0.2 ml, 11.1 mmol), and trifluoroacetic acid (3.0 ml, 39.2 mmol). The suspension quickly clarified and turned yellow upon addition of trifluoroacetic acid. The reaction mixture was vigorously stirred at room temperature for 30 min and then the solvent was removed by rotary evaporation. The resulting colorless oil was triturated with diethyl ether. After decanting the ether washes, residual solvent was removed under vacuum to afford 499 mg (96%) of 2-(4-chlorophenoxy)-N-[(1r,4r)-4-aminocyclohexyl]acetamide trifluoroacetic acid as a white solid. $^1$H NMR (400 MHz, DMSO-$d_6$) δ 7.95 (d, $J$ = 7.87 Hz, 1H), 7.77 (br. s., 3H), 7.31 (d, $J$ = 8.97 Hz, 2H), 6.93 (d, $J$ = 8.97 Hz, 2H), 4.43 (s, 2H), 3.54 (m, 1H), 2.93 (br. s., 1H), 1.90 (d, $J$ = 9.16 Hz, 2H), 1.77 (d, $J$ = 9.34 Hz, 2H), 1.31 (sxt, $J$ = 11.50 Hz, 4H); LC-MS: $m/z$ = 283 [M + H, $^{35}$Cl ]$^+$, 285 [M + H, $^{37}$Cl]$^+$.

Step 3: To a solution of 4-fluorophenoxyacetic acid (0.009 g, 0.050 mmol) in N,N-dimethylformamide (1.0 ml) were sequentially added 1-hydroxybenzotriazole hydrate (0.009 g, 0.055 mmol), 1-(3-dimethylaminopropyl)-3-ethylcarbodiimide hydrochloride (0.012 g, 0.057 mmol), 2-(4-chlorophenoxy)-N-[(1r,4r)-4-aminocyclohexyl]acetamide trifluoroacetic acid (0.02 g, 0.050 mmol) and N,N-diisopropylethylamine (0.013 ml, 0.12 mmol). The reaction mixture was subjected to conditions described in procedure A to obtain 14 mg (60%) of the title compound as a white solid. $^1$H NMR (400 MHz, DMSO-d6) δ 7.88–7.92 (M, 2H), 7.31 (d, J = 9 Hz, 2H), 7.10 (t, J = 8.8 Hz, 2H), 6.92–6.95 (m, 4H), 4.39–4.42 (m, 4H), 3.57 (br. s, 2H), 1.74 ( d, J = 5.9 Hz, 4H), 1.29–1.33 (m, 4H) LC-MS: m/z = 435 [M + H, 35Cl ]+, 437 [M + H, 37Cl]+.

## 2-(4-Fluorophenoxy)-N-[(1r,4r)-4-[2-(4-fluorophenoxy)acetamido]cyclohexyl]acetamide (ISRIB-A9)

To a solution 4-fluorophenoxyacetic acid (0.12 g, 0.7 mmol) in N,N-dimethylformamide (1.0 ml) were sequentially added 1-hydroxybenzotriazole hydrate (0.094 g, 0.7 mmol), 1-(3-dimethylaminopropyl)-3-ethylcarbodiimide hydrochloride (0.140 g, 0.7 mmol), (1r,4r)-cyclohexane-1,4-diamine (0.040 g, 0.35 mmol) and N,N-diisopropylethylamine (0.372 ml, 2.1 mmol). The reaction mixture was subjected to conditions described in procedure B to afford 73 mg (50%) of the title compound. $^1$H NMR (300 MHz, CDCl$_3$) δ 7.02 (t, $J$ = 8.3 Hz, 4H), 6.89–6.90 (m, 4H), 6.38 (d, $J$ = 7.5 Hz, 2H), 4.43 (s, 4H), 3.88 (br. s, 2H), 2.07 (d, $J$ = 5.7 Hz, 4H), 1.36–1.39 (m, 4H) LC-MS: $m/z$ = 419 [M + H]$^+$.

## 2-(4-Methylphenoxy)-N-[(1r,4r)-4-[2-(4-chlorophenoxy)acetamido]cyclohexyl]acetamide (ISRIB-A10)

To a solution 4-methyl-phenoxyacetic acid (0.016 g, 0.101 mmol) in N,N-dimethylformamide (1.0 ml) were sequentially added 1-hydroxybenzotriazole hydrate (0.014 g, 0.101 mmol), 1-(3-dimethylaminopropyl)-3-ethylcarbodiimide hydrochloride (0.02 g, 0.101 mmol), 2-(4-chlorophenoxy)-N-[(1r,4r)-4-aminocyclohexyl] acetamide trifluoroacetic acid (0.04 g, 0.101 mmol) and N,N-diisopropylethylamine (0.06 ml, 0.303 mmol). The reaction mixture was subjected to conditions described in procedure A to obtain 7 mg (16%) of the title compound as a white solid. $^1$H NMR (400 MHz, DMSO-d6) δ 7.91 (d, J = 8 Hz, 1H), 7.84 (d, J = 7.8 Hz, 1H ), 7.31 (d, J = 8.8 Hz, 2H), 7.06 (t, J = 8.3 Hz, 2H), 6.94 (d, J = 8.8 Hz, 2H), 6.80 (d, J = 8.4 Hz, 2H ), 4.42 (s, 2H), 4.35 (s, 2H), 3.56 (br. s, 2H), 2.20 (s, 3H), 1.73 ( d, J = 6.6 Hz, 4H), 1.22–1.33 (m, 4H) LC-MS: m/z = 431 [M + H]$^+$.

## 2-(4-Methylphenoxy)-N-[(1r,4r)-4-[2-(4-methylphenoxy)acetamido]cyclohexyl]acetamide (ISRIB-A11)

To a solution 4-methylphenoxyacetic acid (0.116 g, 0.7 mmol) in N,N-dimethylformamide (1.0 ml) were sequentially added 1-hydroxybenzotriazole hydrate (0.094 g, 0.7 mmol), 1-(3-dimethylaminopropyl)-3-ethylcarbodiimide hydrochloride (0.14 g, 0.7 mmol), (1r,4r)-cyclohexane-1,4-diamine (0.04 g, 0.35 mmol) and N,N-diisopropylethylamine (0.372 ml, 2.1 mmol). The reaction mixture was stirred at 52°C for 24 hr and then subjected to conditions described in procedure B to afford 84 mg (58%) of the title compound. $^1$H NMR (400 MHz, DMSO-d$_6$) δ 7.84 (d, J = 6.8 Hz, 2H), 7.05 (d, J = 6.8 Hz, 4H), 6.80 (d, J = 6.6 Hz, 4H), 4.35 (s, 4H), 3.56 (br. s, 2H), 2.19 (s, 6H), 1.73 (br. s, 4H), 1.31 (br.s, 4H) LC-MS: m/z = 411 [M + H]$^+$.

## 2-(4-Cyanophenoxy)-N-[(1r,4r)-4-[2-(4-chlorophenoxy)acetamido]cyclohexyl]acetamide (ISRIB-A12)

To a solution 4-cyanophenoxyacetic acid (0.009 g, 0.050 mmol) in N,N-dimethylformamide (1.0 ml) were sequentially added 1-hydroxybenzotriazole hydrate (0.009 g, 0.055 mmol), 1-(3-dimethylaminopropyl)-3-ethylcarbodiimide hydrochloride (0.012 g, 0.057 mmol), 2-(4-chlorophenoxy)-N-[(1r,4r)-4-aminocyclohexyl]acetamide trifluoroacetic acid (0.02 g, 0.050 mmol) and N,N-diisopropylethylamine (0.013 ml, 0.12 mmol). The reaction mixture was subjected to conditions described in procedure A to obtain 14 mg (65%) of the title compound as a beige solid. 1H NMR (400 MHz, DMSO-d6) δ 7.99 (d, J = 7.9 Hz, 1H), 7.91 (d, J = 8.1 Hz, 1H), 7.76 (d, J = 8.8 Hz, 1H),

7.31 (d, J = 9.1 Hz, 1H), 7.07 (d, J = 8.8 Hz, 2H), 6.94 (d, J = 8.8 Hz, 2H), 4.55 (s, 2H), 4.42 (s, 2H), 3.56 (br. s, 2H), 1.74 ( d, J = 7.7 Hz, 4H), 1.28–1.32 (m, 4H) LC-MS: m/z = 442 [M + H, 35Cl ]+, 444 [M + H, 37Cl]+.

## 2-(4-Cyanophenoxy)-N-[(1r,4r)-4-[2-(4-cyanophenoxy)acetamido]cyclohexyl]acetamide (ISRIB-A13)

To a solution 4-cyanophenoxyacetic acid (0.124 g, 0.7 mmol) in N,N-dimethylformamide (1.0 ml) were sequentially added 1-hydroxybenzotriazole hydrate (0.094 g, 0.7 mmol), 1-(3-dimethylaminopropyl)-3-ethylcarbodiimide hydrochloride (0.14 g, 0.7 mmol), (1r,4r)-cyclohexane-1,4-diamine (0.04 g, 0.35 mmol) and N,N-diisopropylethylamine (0.372 ml, 2.1 mmol). The reaction mixture was subjected to conditions described in procedure B to afford 54 mg (36%) of the title compound. [1]H NMR (300 MHz, DMSO-$d_6$) δ 8.01 (d, J = 5.8 Hz, 2H), 7.76 (d, J = 6.8 Hz, 4H), 7.08 (d, J = 6.8 Hz, 4H), 4.55 (s, 4H), 3.56 (br. s, 2H), 1.75 (br. s, 4H), 1.31 (br. s, 4H) LC-MS: m/z = 433 [M + H]+.

## 2-(3,4-Dichlorophenoxy)-N-[(1r,4r)-4-[2-(4-chlorophenoxy)acetamido]cyclohexyl]acetamide (ISRIB-A14)

To a solution 3,4-dichlorophenoxyacetic acid (0.011 g, 0.050 mmol) in N,N-dimethylformamide (1.0 ml) were sequentially added 1-hydroxybenzotriazole hydrate (0.009 g, 0.055 mmol), 1-(3-dimethylaminopropyl)-3-ethylcarbodiimide hydrochloride (0.012 g, 0.057 mmol), 2-(4-chlorophenoxy)-N-[(1r,4r)-4-aminocyclohexyl]acetamide trifluoroacetic acid (0.020 g, 0.050 mmol) and N,N-diisopropylethylamine (0.013 ml, 0.12 mmol). The reaction mixture was subjected to conditions described in procedure A to obtain 21 mg (86%) of the title compound as a white solid. [1]H NMR (400 MHz, DMSO-$d_6$) δ 7.94 (d, J = 8.2 Hz, 1H), 7.91 (d, J = 8.2 Hz, 1H), 7.51 (d, J = 8.8 Hz, 1H), 7.31 (d, J = 9 Hz, 2H), 7.22 (d, J = 2.9 Hz, 1H), 6.92–6.95 (m, 3H), 4.48 (s, 2H), 4.42 (s, 2H), 3.56 (br. s, 2H), 1.74 (d, J = 6 Hz, 4H), 1.26–1.31 (m, 4H) LC-MS: m/z = 485 [M + H, 35Cl ]+, 487 [M + H, 37Cl]+.

## 2-(3,4-Dichlorophenoxy)-N-[(1r,4r)-4-[2-(3,4-dichlorophenoxy)acetamido]cyclohexyl]acetamide (ISRIB-A15)

To a solution of (1r,4r)-cyclohexane-1,4-diamine (0.025 g, 0.2 mmol) in N,N-dimethylformamide (1 ml) were added 3,4-dichlorophenoxyacetic acid (0.097 g, 0.4 mmol), 1-[bis(dimethylamino)methylene]-1H-1,2,3-triazolo[4,5-b]pyridinium 3-oxid hexafluorophosphate (0.175 g, 0.5 mmol) and N,N-diisopropylethylamine (0.153 ml, 0.9 mmol). The reaction mixture was subjected to conditions described in procedure C to obtain 107 mg (94%) of the title compound as a cream colored solid. $^1$H NMR (400 MHz, CDCl$_3$) δ 7.37 (d, J = 8.8 Hz, 2H), 7.04 (s, 2H), 6.78 (d, J = 8.8 Hz, 2H), 6.26 (d, J = 8.1 Hz, 2H),4.42 (s, 4H), 3.85 (br. s, 2H), 2.05 (d, J = 6 Hz, 4H), 1.31–1.39 (m, 4H); LC-MS: m/z = 519 [M + H, $^{35}$Cl]$^+$, 521 [M + H, $^{37}$Cl]$^+$.

## 2-(4-Chloro-3-fluorophenoxy)-N-[(1r,4r)-4-[2-(4-chlorophenoxy) acetamido]cyclohexyl]acetamide (ISRIB-A16)

Step 1: To a cooled solution (0°C) of (1r,4r)-4-[2-(4-chlorophenoxy)acetamido]cyclohexan-1-aminium trifluoroacetate (0.550 g, 1.4 mmol) in THF and N,N-diisopropylethylamine (0.966 ml, 5.5 mmol) slowly added chloroacetyl chloride (0.121 ml, 1.5 mmol). The mixture was stirred at ambient temperature for 20 min. The reaction mixture was diluted in dichloromethane, washed with 0.1 N hydrochloric acid, water and brine. The organic layer was dried over magnesium sulfate, filtered and concentrated in a rotary evaporator to obtain about 430 mg of crude 2-(4-chlorophenoxy)-N-[(1r,4r)-4-(2-chloroacetamido)cyclohexyl]acetamide that was used without further purification.

Step 2: To a suspension of 2-(4-chlorophenoxy)-N-[(1r,4r)-4-(2-chloroacetamido)cyclohexyl] acetamide (0.036 g, 0.1 mmol) and 4-chloro-3-fluorophenol (0.015 g, 0.1 mmol) in acetone (1.0 ml), added potassium carbonate (0.021 g, 0.2 mmol) and stirred at 120°C in the microwave reactor for 20 min. The reaction mixture was concentrated down and suspended in water (10 ml). The mixture was vigorously vortexed then centrifuged, and the water was decanted. This washing protocol was repeated with water and then with diethyl ether (10 ml). The wet solid was dissolved in dichloromethane (10 ml) and dried over anhydrous magnesium sulfate. The solids were removed by filtration, and the filtrate was concentrated by rotary evaporation to afford 28 mg (60%) of the title compound as a tan solid. $^1$H NMR (400 MHz, DMSO-d$_6$) δ 7.9 (t, J = 8.9 Hz, 2H), 7.46 (t, J = 8.9 Hz, 1H), 7.31 (d, J = 9 Hz, 2H), 7.03 (dd, J = 11.4, 2.7 Hz, 1H), 6.94 (d, J = 9 Hz, 2H), 6.81 (dd, J = 8.5, 2.3 Hz, 1H), 4.46 (s, 2H), 4.42 (s, 2H), 1.74 ( d, J = 6.2 Hz, 4H), 1.29–1.35(m, 4H) LC-MS: m/z = 469 [M + H, $^{35}$Cl ]$^+$, 471 [M + H, $^{37}$Cl]$^+$.

## 2-(4-Chloro-3-fluorophenoxy)-N-[(1r,4r)-4-[2-(4-chloro-3-fluorophenoxy) acetamido]cyclohexyl]acetamide (ISRIB-A17)

Step 1: To a solution 4-chloro-3-fluorophenol (0.100 g, 0.7 mmol) in N,N-dimethylformamide (2 ml), were added potassium carbonate (0.189 g, 1.4 mmol) and tert-butyl bromoacetate (0.111 ml, 0.8 mmol) and stirred at 65°C for 2 hr. The reaction mixture was diluted with ethyl acetate (10 ml), washed with water (3 × 10 ml) and brine (10 ml). The organic layer was dried over magnesium sulfate and concentrated in a rotary evaporator to obtain 177 mg of tert-butyl 2-(4-chloro-3-fluorophenoxy)acetate as a colorless oil which was used without further purification.

Step 2: To a solution of tert-butyl 2-(4-chloro-3-fluorophenoxy)acetate (177 mg, 0.7 mmol) in methanol/water (4.5 ml, 2:1) was added aqueous 5 N NaOH solution (0.7 ml, 3.5 mmol) and stirred at ambient temperature for an hour. The reaction mixture was concentrated in a rotary evaporator to remove methanol, diluted with water (5 ml) and extracted with ethyl acetate (5 ml). The aqueous layer was adjusted to about pH 2 with 1 N hydrochloric acid and extracted with ethyl acetate (3 × 5 ml). The organic extract was washed with brine (5 ml), dried over magnesium sulfate and concentrated to obtain 108 mg of 2-(4-chloro-3-fluorophenoxy)acetic acid as a white solid which was used without further purification.

Step 3: To a solution of (1r,4r)-cyclohexane-1,4-diamine (0.02 g, 0.2 mmol) in N,N-dimethylformamide (1 ml) were added 2-(4-chloro-3-fluorophenoxy)acetic acid (0.072 g, 0.4 mmol), 1-[bis(dimethylamino)methylene]-1H-1,2,3-triazolo[4,5-b]pyridinium 3-oxid hexafluoro-phosphate (0.14 g, 0.4 mmol) and N,N-diisopropylethylamine (0.122 ml, 0.7 mmol). The reaction mixture was subjected to conditions described in procedure C to obtain 85 mg (>95%) of the title compound as a white solid. $^1$H NMR (400 MHz, DMSO-$d_6$) $\delta$ 7.23–7.28 (m, 2H), 6.72 (d, $J = 8$ Hz, 2H), 6.61–6.64 (m, 4H), 4.36 (s, 4H), 3.56 (m, 2H), 1.95 (d, $J = 6.2$ Hz, 4H), 1.28–1.33 (m, 4H); LC-MS: $m/z = 487$ [M + H, $^{35}$Cl ]$^+$, 489 [M + H , $^{37}$Cl ]$^+$.

## 3-(4-Chlorophenyl)-N-[(1r,4r)-4-[3-(4-chlorophenyl)propanamido]cyclohexyl]propanamide (ISRIB-A18)

To a solution 3-(4-chlorophenyl)propionic acid (0.129 g, 0.7 mmol) in N,N-dimethylformamide (1.0 ml) were sequentially added 1-hydroxybenzotriazole hydrate (0.094 g, 0.7 mmol), 1-(3-dimethylaminopropyl)-3-ethylcarbodiimide hydrochloride (0.14 g, 0.7 mmol), (1r,4r)-cyclohexane-1,4-diamine (0.04 g, 0.35 mmol) and N,N-diisopropylethylamine (0.372 ml, 2.1 mmol). The reaction mixture was stirred at 52°C for 18 hr and then subjected to conditions described in procedure B to afford 103 mg (66%) of the title compound. $^1$H NMR (400 MHz, DMSO-$d_6$) $\delta$ 7.65 (d, $J = 7.5$ Hz, 2H), 7.28 (d, $J = 8.1$ Hz, 4H), 7.17–7.19 (m, 4H), 3.41 (br.s, 2H), 2.73–2.76 (m, 4H), 2.26–2.30 (m, 4H), 1.66–1.68 (m, 4H), 1.10–1.12 (m, 4H) LC-MS: $m/z = 447$ [M + H, $^{35}$Cl ]$^+$, 449 [M + H, $^{37}$Cl]$^+$.

## Acknowledgements

We are indebted to Dr Shiva Malek (Genentech) for suggesting the use of thermodenaturation to monitor ligand-target engagement. We thank Margaret Elvekrog for her technical advice and Diego Acosta-Alvear for reagents, and Jason Gestwicki for invaluable advice on the ISRIB SAR.

## Additional information

### Competing interests

CS: Inventors on UC patent application PCT/US2014/029568. Title: Modulators of the eIF2a pathway.
BRH: Inventors on UC patent application PCT/US2014/029568. Title: Modulators of the eIF2a

pathway. PV: Inventors on UC patent application PCT/US2014/029568. Title: Modulators of the eIF2a pathway. ARR: Inventors on UC patent application PCT/US2014/029568. Title: Modulators of the eIF2a pathway. PW: Inventors on UC patent application PCT/US2014/029568. Title: Modulators of the eIF2a pathway. The other authors declare that no competing interests exist.

## Funding

| Funder | Author |
| --- | --- |
| Howard Hughes Medical Institute (HHMI) | Carmela Sidrauski, Jordan C Tsai, Martin Kampmann, Aaron S Mendez, Jonathan S Weissman, Peter Walter |

The funder had no role in study design, data collection and interpretation, or the decision to submit the work for publication.

## Author contributions

CS, JCT, MK, Conception and design, Acquisition of data, Analysis and interpretation of data, Drafting or revising the article; BRH, Conception and design, Acquisition of data, Analysis and interpretation of data; PV, PJ, BWN, ELT, EV, Acquisition of data, Analysis and interpretation of data; MS, Acquisition of data, Contributed unpublished essential data or reagents; ASM, Analysis and interpretation of data, Contributed unpublished essential data or reagents; JRJ, Conception and design, Analysis and interpretation of data; NJK, CSF, JSW, Analysis and interpretation of data, Drafting or revising the article; ARR, PW, Conception and design, Analysis and interpretation of data, Drafting or revising the article

## Author ORCIDs

Jordan C Tsai, http://orcid.org/0000-0001-5202-722X
Martin Kampmann, http://orcid.org/0000-0002-3819-7019
Erik Verschueren, http://orcid.org/0000-0001-5842-6344

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
