## [Decision Letter]

Thank you for sending your work entitled “Pharmacological dimerization and activation of the exchange factor eIF2B antagonizes the integrated stress response” for consideration at *eLife*. Your article has been favorably evaluated by Randy Schekman (Senior editor) and three reviewers, one of whom, Jeffery W Kelly, is a member of our Board of Reviewing Editors.

The Reviewing editor and the other reviewers discussed their comments before we reached this decision, and the Reviewing editor has assembled the following comments to help you prepare a revised submission.

This study depicts an impressive addition to the small, but growing list, of well-defined small molecule Proteostasis regulators. Specifically, the manuscript addresses and answers an interesting question in the field-that is the mode of action of ISRIB, the only currently available small molecule to block eIF2∼P regulation of translational initiation. The text and experiments are generally clearly presented and support the conclusions drawn. The study has broad biomedical implications.

The first reviewer request is to better integrate the three parts in the manuscript. It would be nice to discuss the other gene classes that were identified as enhancing or repressing the ISR. Were there any key pathways or processes that were identified in the screen that would provide new ideas about translational control in the ISR?

Could any of the more potent compounds be assessed for their ability to inhibit translation initiation in cell culture? Figure 1 and some later figures include error bars and statistical considerations.

The x-axis scale of Figure 5 differs from the other panels and to the casual reader can be confusing when comparing with Figure 5.

One would like to see the authors document the extent of eIF2B subunit knock-down and, importantly, the consequences of knockdown on translation of the ATF4 reporter and bulk translation initiation in the absence of ER stress and attendant elevated eIF2α phosphorylation, in the presence and absence of ISRIB.

It would be expected that reducing eIF2B levels would induce ATF4 translation in the absence of ER stress, by reducing levels of eIF2-GTP-tRNAi ternary complexes, and it seems likely that the drug would still reduce ATF4 translation by stimulating the GEF activity of the residual eIF2B—the authors' thoughts?

ATF4 reporter expression would remain elevated in the knockdown cells compared to normal cells in response to ISRIB treatment simply because reporter expression would be so high to begin with in the knockdown cells. This would differ from the authors’ interpretation of the data in Figure 1, all collected in the presence of ER stress, that eIF2B knockdown renders cells insensitive to ISRIB because “there is a lower supply of molecules that can be activated”. The authors’ explanation seems to imply that eIF2B cannot be activated by the drug when present below a threshold concentration in the cell, which seems rather unlikely-please comment.

The authors’ screen is based on ATF4 reporter expression, not bulk translation, and low levels of eIF2-GTP-tRNAi ternary complexes induce ATF4 by shifting the probability of reinitiation from inhibitory uORF2 to the ATF4 start codon. Do the authors mean to suggest that with decreased overall translation on eIF4G1 knockdown the demand on ternary complexes is diminished, and that the resulting higher level of ternary complexes would oppose reinitiation at the ATF4 start codon; and that ISRIB would exacerbate this effect by increasing eIF2B activity and elevating ternary complex levels further? This passage needs to be clarified.

---

## [Author Response]

*The first reviewer request is to better integrate the three parts in the manuscript*.

We have carefully reread the transition between the different sections of the paper and found that they work quite well, given the unavoidable diversity of the approaches taken. To address the concern, we have added a statement to the Introduction preparing the reader to the flow of the logic that follows.

*It would be nice to discuss the other gene classes that were identified as enhancing or repressing the ISR*. *Were there any key pathways or processes that were identified in the screen that would provide new ideas about translational control in the ISR?*

We only interrogated a limited shRNA library focused genes with implied functions in proteostasis. Thus we are unable to comprehensively assess the impact of modulation of different pathways on translational control of the ISR. There are no other gene classes that stand out prominently enough to warrant an informed discussion that would enhance the manuscript. We noted that a few components of the oligosaccharyl transferase are included in the gene set that renders cells more resistant to ISRIB, but interpret this observation as resulting from an enhancement of generic ER stress that would elevate eIF2α-phosphorylation further, most likely beyond the threshold that ISRIB can reverse. Similarly, knockdown of HYOU1, encoding the ER chaperone Grp170, increased reporter activation. The data are included in the source files of Figure 1. Given the successful biochemical validation ISRIB effects on eIF2B, we did not follow up on these and other genes that changed the cells’ sensitivity to ISRIB.

*Could any of the more potent compounds be assessed for their ability to inhibit translation initiation in cell culture?*
Figure 1
*and some later figures include error bars and statistical considerations*.

We included error bars in Figure 1 and Figure 5. We added a supplementary figure containing the dose response curves (cell-based assay) of all ISRIB analogs tested in Figure 2 with their corresponding error bars. We have not tested the effects of all newly synthesized analogs on bulk translation. ISRIB and other analogs (not described in this paper) that we did analyze, consistently demonstrated that translation upregulation of the uORFs-ATF4 reporter inversely correlates with downregulation of bulk mRNA translation. Both of these outputs depend directly on the ternary complex concentration in the cell; lower levels of ternary complex result in increased uORFs-ATF4-GFP expression and reduced bulk protein synthesis. As all analogs are based on the same scaffold, it is highly unlikely that changes in the molecule would uncouple the effects observed with the ISR-translation reporter from the inversely correlated effects on bulk protein synthesis.

*The x-axis scale of*
Figure 5
*differs from the other panels and to the casual reader can be confusing when comparing with*
Figure 5.

We modified the x-axis of Figure 5 to match the other panels in the Figure.

*One would like to see the authors document the extent of eIF2B subunit knock-down and, importantly, the consequences of knockdown on translation of the ATF4 reporter and bulk translation initiation in the absence of ER stress and attendant elevated eIF2α phosphorylation, in the presence and absence of ISRIB*.

We have not measured the degree of shRNA knockdown. Since the screen deployed a pool of 25 different shRNAs representing every gene, the degree of knockdown is different from one shRNA to another. This creates an allelic series of different knockdown efficiencies that are integrated in the analysis to compute statistical significance plotted as the P-value. This spread of knockdown efficiencies provides the basis of the screen design as it buffers from lethal effect of complete disruption of gene expression. As such, measurement of individual knockdown efficiencies would provide no further value in interpreting the screen results.

*It would be expected that reducing eIF2B levels would induce ATF4 translation in the absence of ER stress, by reducing levels of eIF2-GTP-tRNAi ternary complexes, and it seems likely that the drug would still reduce ATF4 translation by stimulating the GEF activity of the residual eIF2B*—*the authors' thoughts?*

We agree. This is consistent with our model. In the reviewer’s scenario there is no phosphorylated eIF2α. If ISRIB enhances the intrinsic GEF activity of eIF2B, then we would expect that ISRIB would reduce ATF4 translation in eIF2B-knockdown cell in the absence of stress.

*ATF4 reporter expression would remain elevated in the knockdown cells compared to normal cells in response to ISRIB treatment simply because reporter expression would be so high to begin with in the knockdown cells. This would differ from the authors’ interpretation of the data in*
Figure 1*, all collected in the presence of ER stress, that eIF2B knockdown renders cells insensitive to ISRIB because* “*there is a lower supply of molecules that can be activated*”*. The authors’ explanation seems to imply that eIF2B cannot be activated by the drug when present below a threshold concentration in the cell, which seems rather unlikely-please comment*.

Although the alternative interpretation of the effects of eIF2B knockdown presented by the reviewer is a formal possibility, we believe that the direct evidence presented later in the paper (demonstrating that ISRIB dimerizes and activates of eIF2B) strongly support our proposed interpretation that eIF2B knockdown renders cells insensitive to ISRIB because there is a lower supply of molecules that can be activated. The ratio of eIF2α-P:eIF2B was previously shown to be a critical parameter for ISR induction, explaining how a limited amount of eIF2α-P blocks translation disproportionately. Under reduced eIF2B levels, ISRIB cannot compensate for the inhibitory effects of this phosphorylation event (as shown in Figure 5).

*The authors’ screen is based on ATF4 reporter expression, not bulk translation, and low levels of eIF2-GTP-tRNAi ternary complexes induce ATF4 by shifting the probability of reinitiation from inhibitory uORF2 to the ATF4 start codon. Do the authors mean to suggest that with decreased overall translation on eIF4G1 knockdown the demand on ternary complexes is diminished, and that the resulting higher level of ternary complexes would oppose reinitiation at the ATF4 start codon; and that ISRIB would exacerbate this effect by increasing eIF2B activity and elevating ternary complex levels further? This passage needs to be clarified*.

Yes, this is what we think. ISRIB exacerbates diminished initiation at the ATF4 start codon by increasing eIF2B activity and elevating ternary complex levels further. As reported in our 2013 *eLife* paper (Figure 3–figure supplement 4), ISRIB increased 43S loading on mRNAs as shown by the persistence of halfmers in the monosome and disome peaks in the polyribosome analysis of cells treated with ER-stressors and ISRIB. These data provide strong support to the notion that ISRIB restores ternary complex levels in ER stressed cells.